# Snow water equivalent retrieved from X- and dual Ku-band scatterometer measurements at Sodankylä using the Markov Chain Monte Carlo method

Jinmei Pan[1], Michael Durand[2], Juha Lemmetyinen[3] , Desheng Liu[4], Jiancheng Shi[1]

[1]National Space Science Center, Chinese Academy of Sciences, Beijing 100190, China
[2]School of Earth Science and Byrd Polar Research Center, The Ohio State University, Columbus, OH 43210, USA
[3]Finnish Meteorological Institute, Helsinki FI-00101, Finland
[4]Department of Geography, The Ohio State University, Columbus, OH 43210, USA

*Correspondence to*: Jinmei Pan (jinmei.pan@gmail.com; panjinmei@nssc.ac.cn)

**Abstract.** Radar at high frequency is a promising technique for fine-resolution snow water equivalent (SWE) mapping. In this paper, we extend the Bayesian-based Algorithm for SWE Estimation (BASE) from passive to active microwave (AM) application and test it using ground-based backscattering measurements at three frequencies (X- and dual Ku-bands, 10.2, 13.3 and 16.7 GHz), VV polarization obtained at 50° incidence angle from the Nordic Snow Radar Experiment (NoSREx) in Sodankylä, Finland. We assumed only an uninformative prior for snow microstructure, in contrast with an accurate prior required in previous studies. Starting from a biased monthly SWE prior from land surface model simulation, two-layer snow state variables and single-layer soil variables were iterated until their posterior distribution could stably reproduce the observed microwave signals. The observation model is the Microwave Emission Model of Layered Snowpacks 3 and Active (MEMLS3&a) based on the Improved Born Approximation (IBA). Results show that BASE-AM achieved a RMSE of ~10 cm for snow depth (SD) and less than 30 mm for SWE, compared with the RMSE of ~20 cm SD and ~50 mm SWE from priors. Retrieval errors were significantly larger when BASE-AM was run using a single snow layer. The results support the potential of X- and Ku-band radar for SWE retrieval and show that the role of a precise snow microstructure prior in SWE retrieval may be substituted by a SWE prior from exterior sources.

## 1 Introduction

Every year, snow and ice covers about 50% of the land surface in the Northern Hemisphere (Brown & Robinson, 2011), reflects back up to 80% of the solar radiation, cools the Earth's surface (Flanner et al., 2011), and provides water supplies for about 1/6 of the world's population (Barnett et al., 2005). The estimation of snow water equivalent (SWE), which describes the equivalent depth of liquid water when snow completely melts (Takala et al., 2011), is of critical importance for hydraulic and hydrological applications (Lettenmaier et al., 2015). However, current observation-based estimates of global snow lack precision and spatial resolution needed to capture global processes (Mortimer et al., 2020); snow is the most poorly measured component of the global water cycle (Durand et al., 2021).

Active microwave radar at the X- and Ku-bands shows great promise for high resolution snow depth and SWE mapping (Rott et al., 2012; Tsang et al., 2022). This technique is based on detecting changes in volume scattering from the snow

medium, and thus builds from the heritage from passive microwave remote sensing (Tsang et al., 2022). Active microwave remote sensing can achieve far higher spatial resolution than passive microwave via synthetic aperture radar (SAR) processing. The existence of snow on the ground and its volume scattering generally increases the backscattered radar signal

as compared to bare soil. Multiple satellite mission proposals have proposed to use this technique, but none have so far been selected for space-borne operations. The Snow and Cold Lands Processes (SCLP) mission proposed to NASA, the Cold Regions Hydrology High-resolution  Observatory (CoReH2O) proposed to ESA, and The Water Cycle Observation Mission (WCOM) proposed in China were all to have been dual-frequency radars operating at X- and Ku- bands (Cline et al., 2007; Rott et al., 2012; Shi et al., 2014). The Canadian Space Agency (CSA) is currently considering a concept study for a satellite

radar mission for terrestrial snow mass, proposing a dual Ku-band scatterometer (Derksen et al., 2021; Tsang et al., 2022). The maturity of the algorithms to retrieve SWE from radar signal has grown significantly as described by Tsang et al. (2022), but algorithm challenges remain. New work on Ku- and X- retrievals is continuing, for example in the NASA SnowEx experiments featuring the SWESARR instrument (Rincon et al., 2020).

Algorithm development for Ku- and X- SAR retrievals is of vital importance. The radar backscatter from snow is sensitive to

SWE but is complicated by confounding factors including snow microstructure, the backscatter from the substrate beneath snow, and forest cover (Tsang et al., 2022). Recent advances have begun to resolve the substrate issue, specifically by subtracting the contribution of the rough surface scattering at the snow-soil interface (e.g. Zhu et al., 2018) and using passive microwave measurements (Zhu et al., 2021). Forests pose an important limitation on the applicability of the technique, and recent studies have helped refine estimates of forest conditions under which SWE may be estimated (e.g. Macelloni et al.,

2017; Lemmetyinen et al., 2022). In this paper, we focus on retrieval issues posed by snow microstructure.

The complexities of retrieving SWE from radar measurements derive from fundamentals of snow physics and electromagnetic physics. Radar backscatter is highly sensitive to snow microstructure, commonly characterized by the size of the individual snow crystals (e.g. Xu et al., 2010; King et al., 2018; Rutter et al., 2019). Because grain shape is highly irregular and exhibits significant spatiotemporal variability, and because grains are oftentimes well-bonded within snowpack,

we often refer to "snow microstructure" or to the microstructure correlation length rather than grain size (Picard et al., 2023). The snow correlation length can considered as the length scale describing the auto-correlation function (ACF) of the ice-air medium, signifying the distance within which this medium can still be considered correlated (Mätzler, 1997). In this study, we specifically estimate the exponential correlation length of the snow microstructure. The distinction lies in how the correlation length is determined. While correlation length is fitted from the ACF near the origin, the exponential correlation

length is fitted from a longer range of two-point distances in the medium (Mätzler, 2002). High values of correlation lengths generally correspond to high values of grain size; Pan et al. (2017) explored the relationship between grain size and correlation length for the NoSREx dataset. Radar backscatter is quite sensitive to snow microstructure, and the dependence is highly non-linear. These complexities have led algorithm developers to introduce a priori information on grain size to help constrain the retrieval problem (Tsang et al., 2022), which in turn makes the retrieved SWE accuracy dependent on the

unbiasedness of the prior grain size, at least to some extent. The CoReH2O mission specified that an effective grain radius

must be known a priori to within 15% precision to enable SWE retrieval, a daunting requirement indeed. Rutter et al. (2019) similarly found in context of a sensitivity experiment that depth hoar equivalent grain size must be specified to within 5-10% precision in order to achieve a $\pm 30$ mm SWE accuracy requirement (IGOS, 2007). Despite recent advances, capabilities to predict grain size still fall below the required precision, creating a dilemma for retrieval algorithms. Lemmetyinen et al.

(2018) demonstrated that radar SWE retrieval can be supported by scaling the effective correlation length obtained from passive microwave observations, which, however, was fitted using snow depth measurements, too. As described by Tsang et al. (2022), the approach of Cui et al. (2016) and Zhu et al. (2018; 2021) reframes the need for priori information, by changing it from a snow microstructure parameter to the single-scattering albedo at X-band. Furthermore, Merkouriadi et al. (2021) indicated that applying physical model to generate priors of grain size is not straightforward, as biased SWE

estimates in physical models lead to large biases in the modelled microstructure, which in turn propagate as an increase of bias in a potential microwave retrieval using these as priors.

While past work focused on an error propagation approach to infer the required precision for an equivalent grain size, retrieval algorithms that explicitly statistically model each unknown term in the retrieval problem have not been explored in the literature. Here, we extended the Bayesian-based Algorithm for SWE Estimation (BASE) (Pan et al., 2017) to active

microwave (AM) application. Unlike a simple steepest descent algorithm or Newton's method, a Markov Chain Monte Carlo (MCMC) method is used in BASE-AM, providing posterior distributions of several variables at the same time from observations and prior distributions, without any assumption of linear error propagation. The MCMC method looks for a global optimization instead of a local optimization. We tested BASE-AM SWE retrieval using ground-based radar measurements from the Nordic Snow Radar Experiment (NoSREx; Lemmetyinen et al., 2016a). We used the Microwave

Emission Model of Layered Snowpacks 3 and Active (MEMLS3&a) (Proksch et al., 2015) based on the Improved Born Approximation (IBA) as the observation model, and consider a two-layer snow structure, composed of a surface layer and a bottom layer. We iteratively updated the snow (snow layer thickness, exponential correlation length, density and temperature), soil (soil temperature, roughness and total water content), and model variables in MEMLS 3&a to build MCMC chains. The model variable iterated is Q, a semi-empirical parameter to separate the total backscattering into co- and

cross-polarization components. The exponential correlation length is the snow microstructure parameter specifically used in MEMLS 3&a. We deliberately chose a biased monthly SWE prior from land surface model simulations compared with the in-situ observations, and thus implicitly tested whether radar data can overcome such biases. Acknowledging the challenge of obtaining appropriate snow microstructure priors, we chose a fixed and nearly uninformative prior for exponential correlation length, which followed a normal distribution as $N$(0.18 mm, 0.09 mm) for both snow layers, whereas 0.18 mm is

the mean and 0.09 mm is the standard deviation.

If this SWE retrieval algorithm for radar using MCMC approach successfully estimates SWE, then the two-layer approach with a biased prior on SWE and an uninformative prior for snow microstructure contains adequate information to estimate SWE in support by three-frequency radar observations, thus providing a new perspective on the need for priori information outlined by Rott et al. (2012) and Rutter et al (2019). If the algorithm is unsuccessful, we'll have found that a precise prior

for snow microstructure are required, in agreement with previous literature. We hypothesize, based on previous results for passive microwave (Pan et al., 2017) at this site, that the radar retrieval algorithm will also successfully estimate SWE using the same generic prior.

## 2 Data

From 2009 to 2013, continuous snow radar experiments were conducted at the Intensive Observation Area (IOA) (67.362 °N,
26.633 °E) located in the Finnish Meteorological Institute Arctic Research Centre (FMI-ARC) in Sodankylä, Finland, during the NoSREx campaign (Lemmetyinen et al., 2016a). The IOA is located in a clearing of a typical Scots pine (Pinus sylvestris) boreal forest on mineral soil.  A The X- and dual-Ka band scatterometer SnowScat of the European Space Agency (ESA) was installed on a tower at the height of 9.6-m to observe the undisturbed, natural snowpack at several incidence and azimuth angles. Manufactured by GAMMA Remote Sensing, SnowScat,  is a stepped frequency, four-polarization (VV, HH,
VH and HV polarizations) radar operating from 1-18 GHz. The single-look complex measurements were sampled to 1 GHz bands with centre frequencies of 10.2, 13.3 and 16.7 GHz. SnowScat provides an internal calibration loop for tracking stability of the transmitted signal. Stability was further verified to be within the goal of +/- 1 dB by observing an aluminum sphere target before and after each acquisition scan.

The goal of NoSREx was to observe the backscattering coefficient ($\sigma_0$) from before snow onset until after snow
disappearance at regular intervals. For the first season in 2009-2010, a three-hour measurement interval was used. This was extended to four hours for later seasons (2010-2011, 2011-2012, 2012-2013). The seasons are referred to as Intensive Observation Periods (IOPs) 1-4 in this paper. The acquisition scan consisted of 17 independent look directions in azimuth at four elevation angles, corresponding to ground incidence angles of 30, 40, 50 and 60°. Our retrieval used only the VV polarization at a fixed 50° incidence angle. The applied backscattering values represent an average over the 17 independent
looks. We selected this incidence angle to increase the slant penetration path of the radar into the snow medium. However, we avoided choosing an angle that was too large to prevent potential influences from the surrounding environment. We exclusively employed VV polarization, as VH was empirically estimated by MEMLS3&a. It's worth noting that reducing the number of microwave measurements in retrieval typically increases the difficulty of the retrieval process.

Concurrently, snowpits were excavated near the SnowScat, twice per week for the first season and weekly for the following
seasons. Snow depth, snow stratigraphy and geometric grain size of each snow layer were measured; the snow temperature and snow density were measured by 10-cm and 5-cm steps, respectively. There were also several automatic sensors installed at IOA to provide additional information. Continuous SWE measurements were available from the Gamma Snow Instrument (GWI), although it tends to be contaminated by high-frequency noise at short time scales. Soil temperature and soil liquid water content were measured by the Delta-T Devices ML2x sensor at 2-cm in four IOPs, and by the Decagon 5TM sensors at
5, 10, 20, 40 and 80 cm in IOP3 and 4. Soil at the IOA has a texture of 70% sand, 29% silt and 1% clay, and a bulk density of 1300 kg/m$^3$ (Lemmetyinen et al., 2016a). Mineral soils are overlain by a thin organic layer of 2-5 cm of lichen and heather.

Fig.1 shows the measured backscattering coefficients at VV-pol. at 50° incidence angle, with the measured SWE, geometric grain size ($D_g$), snow density and soil liquid water content. The $D_g$ and snow density presented here are mass-weighted average values along the snow profile. From the figure, there is not a simple relationship between volume scattering and maximum SWE among the four IOPs. For example, IOP2 has the maximum backscatter values at 16.7 GHz and the maximum ratio of the 16.7 and 10.2 GHz channels; however, the maximum SWE for IOP2 is the smallest of the four years. At the same time, IOP2 has a relatively high geometric snow grain size ($D_g$), larger than IOP3 and 4 and comparable to IOP1. It agrees with the physical theory that a shallower snow tends to have a larger snow grain size, because of a stronger temperature gradient inside the snow (Jordan, 1991). Thus, among the four IOPs, the influence of snow microstructure is higher than that of SWE. However, as shown in Fig.2, within each IOP, the backscattering ratio between 16.7 and 10.2 GHz increases with both snow depth (SD) and geometric snow grain size ($D_g$), and the relationship have significant differences in different IOPs. During IOP2 and 3, the correlations between the backscattering ratio and SD in (a) can be classified into two groups; this is because of a stronger snow grain growth speed at the very early snow season compared to the mid-snow season.

Concerning other variables, Fig.1 shows the IOPs with a higher maximum SWE tends to have a higher average snow density. It also agrees with the snow compaction theory according to snow process model (Jordan, 1991). The underlying soil was frozen for most of the snow season; however, it was unfrozen and had a large soil water content in the early and late snow season. In almost all IOPs, we observed a decreasing trend of backscattering at all frequencies at the beginning of snow season caused by soil freezing processes. Fig.2(c)-(d) shows more details with respect to the measured backscattering coefficient at 10.2 GHz. Specifically, IOPs with a lower liquid water content or a higher average snow density tends to have higher backscattering at 10.2 GHz. The significantly different $\sigma_0^{VV}$ at low frequency between different IOPs and its seasonal variation indicate a requirement to estimate soil and snow parameters simultaneously.

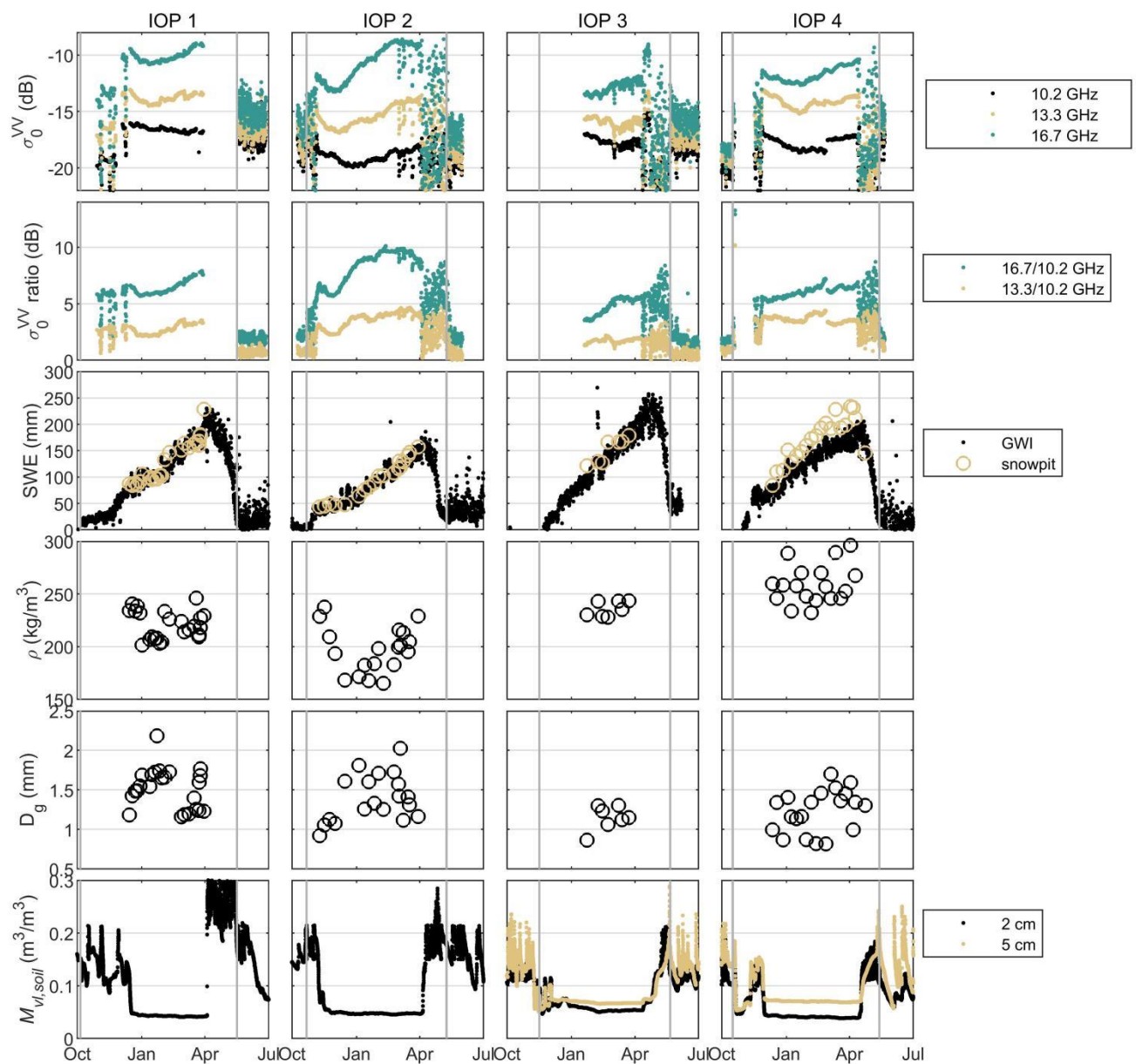

Figure 1: Measured backscattering coefficient at VV-pol. ($\sigma_0^{VV}$) and snow parameters in four IOPs in NoSREx in different columns. The first row shows $\sigma_0^{VV}$ at three frequencies at 50° incidence angle; the second row shows ratio of $\sigma_0^{VV}$ between two frequencies; the third row show the snow water equivalent (SWE); the fourth row shows the profile-average snow density weighted by snow mass ($\rho$); the fifth row shows the profile-average geometric grain size weighted by snow mass ($D_g$); the sixth row shows the measured liquid water content ($M_{vl,soil}$) by Delta-T Devices ML2x sensor (2 cm) and the Decagon 5TM sensors (5 cm). The circles are used for measurements from snowpits for SWE, $\rho$, and $D_g$. Grey vertical bars are used to indicate the onset and end of snow season, identified according to snow depth measurements.

165

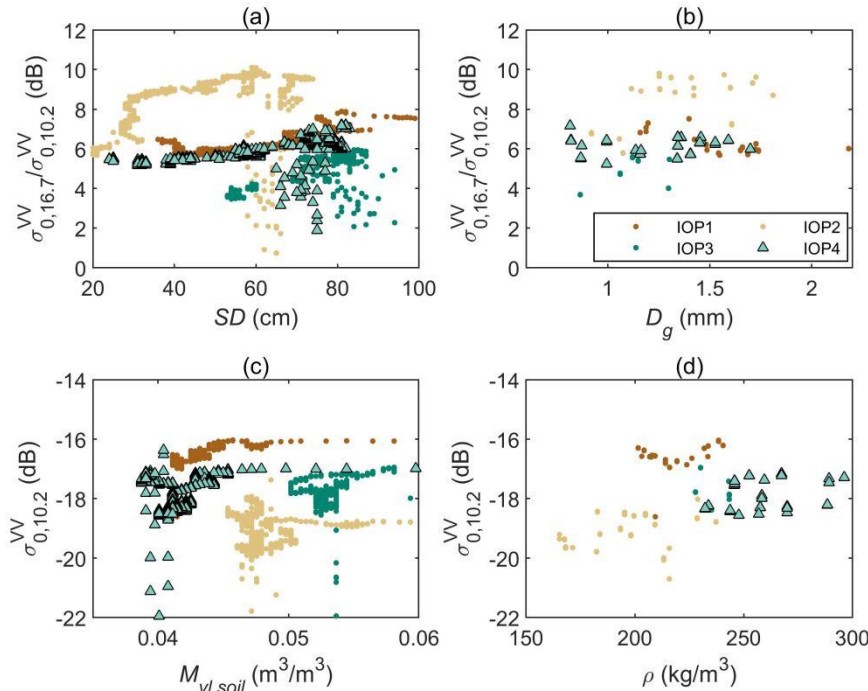

**Figure 2: Relationships between measured ratio of $\sigma_0^{VV}$ between 16.7 and 10.2 GHz and snow depth (*SD*) (a) and geometric grain size ($D_g$) (b), respectively. Relationships between measured $\sigma_0^{VV}$ at 10.2 GHz in frozen soil period with soil liquid water content (c) and snow density (d), respectively. All the data presented here require a soil temperature at 2 cm < 0 °C.**

## 3 Method

### 3.1 The Markov Chain Monte Carlo (MCMC) method

In this paper, we adapted the Bayesian-based Algorithm for SWE Estimation (BASE) used for inverting passive microwave radiometer measurements in Pan et al. (2017) to radar data. The Markov Chain Monte Carlo method (Gelman et al., 1995) is a numerical realization of the Bayes' theorem. Starting from a prior distribution of predicted variables, MCMC randomly searches within the minimal to maximum range of each variable and picks estimates that are both close to the prior and to the observations through the radiative transfer model. The likelihood ratio (R) is used to assess the relative proximity of two sets of SWE and soil parameters to the prior and the observations:

$$R = \frac{P_{obs}(M(x_{i+1}))P_{pr}(x_{i+1})}{P_{obs}(M(x_i))P_{pr}(x_i)} \tag{1}$$

where, $x_i$ is the first set of predicted variables, $x_{i+1}$ is the second set. $P_{pr}(x_i)$ *and* $P_{pr}(x_{i+1})$ are the probability of $x_i$ and $x_{i+1}$ according to the prior distribution. M is the MEMLS3&a observation model, and $P_{obs}(M(x_i))$ and $P_{obs}(M(x_{i+1}))$ are the probability of model-simulated backscattering coefficients, $M(x_i)$ and $M(x_{i+1})$, according to a normal distribution centered at

the observed backscattering coefficients, with a standard deviation of 0.5 dB at each frequency and zero covariance between different frequencies.

At each iteration, if the MCMC algorithm tries to change the value of an estimated variable from $x_i$ to $x_{i+1}$, the likelihood ratio, $R$, is calculated. If $R$ is larger than 1 or larger than a uniformly-distributed random threshold $R_c$ between 0 and 1, the change will be accepted; otherwise, it will be rejected. The randomness in $R_c$ is used to prevent local optimization. Finally, all the iterations will build a vector for each estimate variable, which is called the MCMC chain. The MCMC chain is the numerical realization of the posterior distribution, from which we can calculate the final retrieval results and their uncertainties using the mean and the standard deviation.

MCMC applied here differs with Pan et al. (2017) in the following aspects:

1) The observations and observation model were changed from passive to active (radiometer brightness temperature to radar backscatter).

2) We used basically the same generic prior for SWE estimation in Pan et al.(2017). However, the prior distributions were changed from lognormal distributions to normal distributions, because lognormal-distributed prior leads to lognormal-distributed posterior distribution, which has a skewness that is challenging to interpret.

3) Snow layer thicknesses were independently estimated in Pan et al. (2017), whereas in this study, we estimate the bottom-layer thickness and the relative ratio of the surface layer thickness compared to it. This is needed because we found that the radar at these frequencies has a small sensitivity to volume scattering from the surface snow layers of small grain sizes. The use of layer thickness ratio predetermined the existence of the surface layer, and it was assumed to follow $N(1, 0.2)$. In addition, we constrained the density and temperature of the surface layer to be lower than the bottom layer. These constraints were realized by dynamically using the bottom-layer density and temperature to be the upper limits of those in the surface layer at each iteration. They were set according to a taiga snow type, simply for producing a more reasonable profile feature for these two parameters not very sensitive to backscattering observations. Therefore, these constraints can be revised or deleted if detailed prior knowledge is provided.

As a summary, Fig. 3 shows a flowchart of the retrieval algorithm. We refer to this algorithm as BASE-AM (Active Microwave). As in Pan et al. (2017), the algorithm runs 20,000 iterations, with a burn-in period of 5,000. The burn-in period was used to allow the variables to walk from the initial status to a status that can stably reproduce the observation. The final retrieval results were averaged from the MCMC estimates between the 5,001 to 20,000 iterations.

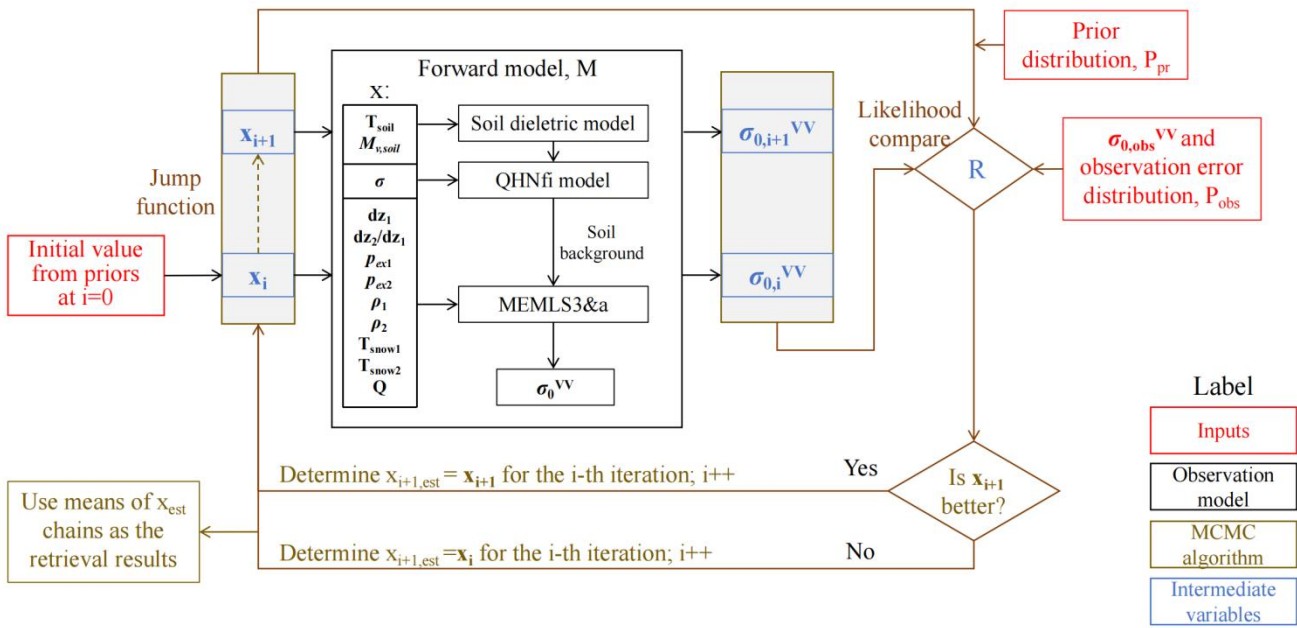

**Figure 3: A flowchat of BASE-AM.**

Table 1 provides a comprehensive list of variables estimated in the MCMC algorithm, along with their respective priors. The
snow and soil priors are generally aligned with the generic prior used in Pan et al. (2017) at the same site. The main
distinction lies in the standard deviations of priors, which were halved compared to Pan et al. (2017). This adjustment was
made because the boundaries of normal distributions are less constrained than lognormal distributions. As in Pan et al.
(2017), the SWE prior was a multiple-year average of monthly-mean SWE from global 2-degree VIC model simulations
(Nijseen et al., 2001). The snow density and snow temperature priors were from the taiga snow type in Sturm's snow classes
(Sturm et al., 1995; 2000). The prior for snow exponential correlation length was uninformative, following $N$(0.18 mm, 0.09
mm). It could be interesting to explore the performance using a different uninformative $p_{ex}$ prior mean, although it is beyond
the scope of this paper. The soil temperature prior was set the same as snow temperature prior. The prior for total soil water
content was set to follow $N$(8% vol, 4% vol). The soil roughness prior was set to follow $N$(1 cm, 0.5 cm).

**Table 1. Summary of priors and boundaries for each estimate variables**

| Parameter | | Mean of prior | Standard deviation (s.t.d.) of prior | Minimal value allowed in MCMC | Maximum value allowed in MCMC | Other constraints |
|---|---|---|---|---|---|---|
| Snow depth | Thickness of the bottom layer ($dz_1$) | Calculated from priors for snow density and SWE from VIC simulations* | Calculated from s.t.d. of snow density and SWE from VIC simulations* | 1 mm | 10 m | No |

| | 1 | 0.2 | 0.001 | 1 | No |
| --- | --- | --- | --- | --- | --- |
| Relative thickness of surface layer compared to bottom layer ($dz_2/dz_1$) | | | | | |
| Snow density ($\rho$) | 217 kg/m3 | 56 kg/m$^3$ | 50 kg/m$^3$ | 917 kg/m$^3$ | Surface density $\leq$ bottom density |
| Exponential correlation length ($p_{ex}$) | 0.18 mm | 0.09 mm | 0.001 mm | 5 mm | No |
| Snow temperature ($T_{snow}$) | -10 ℃ | 5 ℃ | -30 ℃ | 0 ℃ | Surface temperature $\leq$ bottom temperature |
| Soil temperature ($T_{soil}$) | -10 ℃ | 5 ℃ | -30 ℃ | 0 ℃ | No |
| Soil RMSE-height ($\sigma$) | 1 cm | 0.5 cm | 0 cm | 10 cm | No |
| Total soil water content ($M_{v,soil}$) | 8% | 4% | 0% | 100% | No |
| MEMLS3&a polarization splitting parameter ($Q$) | 0.1 | 0.01 | 0.08 | 0.12 | No |

* the mean of SWE prior is VIC-simulated average value of monthly mean SWE, and the s.t.d. is half of the mean.

### 3.2 The MEMLS3&a snow backscattering model

The forward observation model to calculate the snow backscattering was the Microwave Emission Model of Layered Snowpacks 3 and Active (MEMLS 3&a) (Proksch et al., 2015) based on the Improved Born Approximation (IBA). MEMLS3&a for active backscattering simulation is a semi-empirical model that converts passive microwave reflectivity of the snowpack to backscattering. It assumes the distribution of the diffuse part of the bistatic scattering coefficient is Lambertian. In this model, first the snow reflectivity calculated by passive MEMLS ($r$) is separated into a specular part ($r_s$) and a diffuse part ($r_d$). $r_s$ is calculated from the specular part of soil reflectivity, attenuated by snow absorption and snow scattering layer by layer (see equation (14) in Proksch et al. (2015)). Afterward, $r_d=r-r_s$, and $r_d$ is converted to the diffuse part of the backscattering coefficient ($\sigma_d^0$) based on the Lambertian assumption as:

$$\sigma_d^0 = 4r_d\mu_0^2 \tag{2}$$

where. $\mu_0$ is $\cos(\theta_0)$, and $\theta_0$ is the incidence angle.

After $\sigma_d^0$ is calculated, the specular part of backscattering coefficient ($\sigma_s^0$) is calculated using Geometrical-Optics (GO) theory; $\sigma_s^0$ is assumed the same for co-polarizations and zero for cross-polarizations, and was calculate from the specular part of reflectivity ($r_s$) and a roughness parameter ($m^2$) (see equation (9) in Proksch et al. (2015)). Finally, the total snow backscattering coefficient is calculated as:

$$\sigma_{pp'}^0 = \begin{cases} (1-Q)\sigma_{d,v}^0 + \sigma_s^0 & p = p' = v \\ (1-Q)\sigma_{d,h}^0 + \sigma_s^0 & p = p' = h \\ Q\left(\sigma_{d,v}^0 + \sigma_{d,h}^0\right)/2 & p \cong p' \end{cases} \tag{3}$$

In MEMLS3&a, there are two empirical parameters. The first one is $Q$, utilized to split $\sigma_{d,v}^0$ into co-polarizations and cross-polarizations. Our forward simulation test based on snowpit measurements suggested a $Q$ between 0.08 and 0.12 at Sodankylä. Therefore, the prior for $Q$ was set to follow $N(0.1, 0.01)$ in Table 1. The second empirical parameter in MEMLS3&a is the roughness parameter in GO, which is the mean-squared slope ($m^2$). We fixed $m^2$ to a value of 0.01 according to simulations, because we found it only influences backscattering at very small incidence angles ($< 30°$).

### 3.3 Soil models

MEMLS3&a requires the total and specular part of soil reflectivity, instead of soil backscattering. We utilized the QHN model with frequency-independent parameters (QHNfi) developed in Montpetit et al. (2015) to calculate the total soil surface reflectivity. The specular part ($\Gamma_p^s$) of the soil reflectivity is calculated as (Mo et al., 1987; Wegmüller and Matzler, 1999):

$$\Gamma_p^s = \left[(1 - Q_s)\Gamma_p^* + Q_s\Gamma_q^*\right]exp(-4k^2\sigma^2cos^2\theta_1) \tag{4}$$

where, $\Gamma_p^*$ and $\Gamma_q^*$ are the Fresnel reflectivities. $k$ is the wave number. $\sigma$ is the soil roughness (RMS-height of soil surface). $\theta_1$ is the local incidence angle at the snow-soil boundary. $Q_s$ is the polarization mixing parameter for soil surface scattering, which is set the same as in QHNfi.

The soil dielectric constants were calculated using a revised Generalized Refractive Mixing Dielectric Model (GRMDM) (Mironov et al., 2004) adapted for frozen soil. The frozen soil is considered as a mixture of dry solids, bound water, transient water and ice as in Mironov et al. (2017). The model utilized the same bound water content as in Mironov et al. (2004), whereas it fitted temperature-dependent bound water dielectric constants and other important variables using a frozen soil dataset measured at the Beijing Normal University; see Appendix I. The soil texture was set the same as the measurements from NoSREx (Lemmetyinen et al., 2016a).

## 4 Results

### 4.1 MCMC performance for active SWE retrieval

Fig. 4-6 uses the snowpit measured on March 13th, 2012 during IOP 3 (hereafter we call it Pit 49) as an example to show how MCMC works. Fig. 4 shows the simulated backscattering coefficient at each iteration, whereas Fig. 5 and 6 shows the variations of all estimated variables at each iteration.

Fig. 4 shows that the simulated backscattering coefficients at each iteration in the chain (after the burn-in period) are close to the observations, which is one of the key characteristics of MCMC-based retrieval. The mean bias is 0.23, 0.68 and -0.66 dB at 10.2, 13.3 and 16.7 GHz, respectively. The enlarged plot on right shows the influence of initial values of variables on the first 50 iterations.

Fig 5 shows the MCMC chains (left) and the posterior distributions (right) of the snow variables. The posterior distributions of layer thickness (first row) have lower uncertainty than the priors, and the mean is different with the prior. For the estimation of exponential correlation length ($p_{ex}$) (second row), the BASE-AM algorithm predicts a smaller surface-$p_{ex}$ and a larger bottom-layer $p_{ex}$, although the relative relationship between two layers for $p_{ex}$ was not constrained. As shown in the width of the posterior distribution, the uncertainty of surface layer $p_{ex}$ is larger than bottom-layer $p_{ex}$. In the third row, the surface-layer snow density is smaller than bottom-layer density. However, this is simply a realization of our constraint on the relative relationship between two layers. In the fourth row, the backscattering coefficient is not sensitive to snow temperature, so the posterior distributions of snow temperature highly overlap with the priors. The surface-layer snow temperature was constrained to be lower than the bottom layer in MCMC iterations, and it was realized as shown here.

Fig 6 shows the MCMC chains (left) and the posterior distributions (right) of the soil and model variables. The first and fourth rows show that the backscattering coefficient is not sensitive to soil temperature or to the model variable, $Q$: their posterior distributions follow the prior distributions. The second and third rows show that the algorithm gives a small value for total soil water content, whereas the soil roughness varies. This behavior is not identical among snowpits; the total soil water content can have greater variability than soil roughness. The backscattering coefficient at low frequency in this paper is limited, so the algorithm cannot stably estimate soil roughness and soil moisture variables at the same time. This is discussed in detail in Section 5.1.

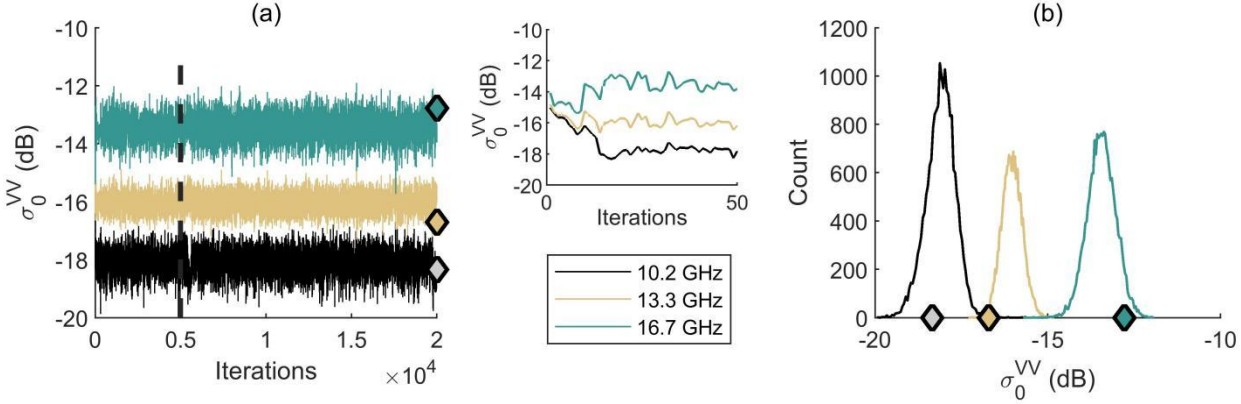

**Figure 4: MCMC chain of simulated backscattering coefficients (lines) compared with the measured backscattering coefficients (diamonds) for Pit 49: (a) in chains of 20,000 iterations with an enlarged plot showing the first 50 iterations on right, (b) in histograms.**

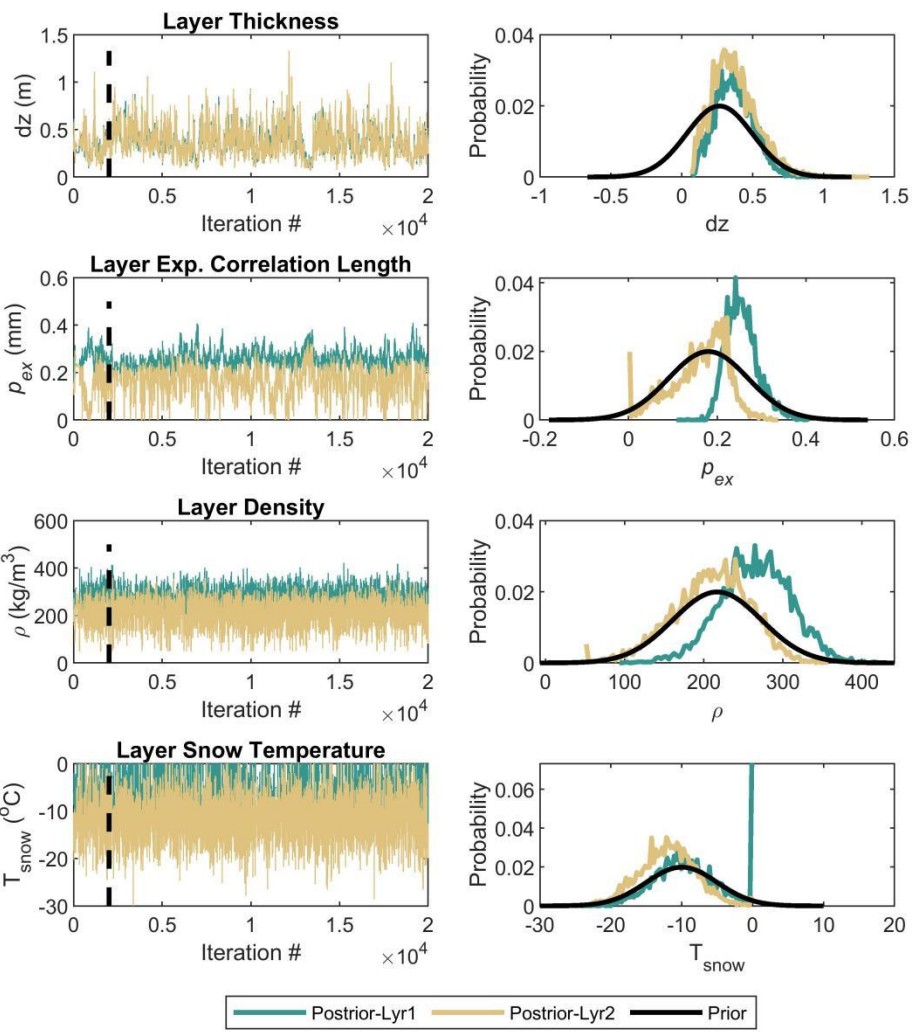

**Figure 5: MCMC chain of layered snow properties (first column) and their posterior distributions compared with the prior distributions (second column) for Pit 49. Lyr1 is for the bottom layer, whereas Lyr2 is for the surface layer.**

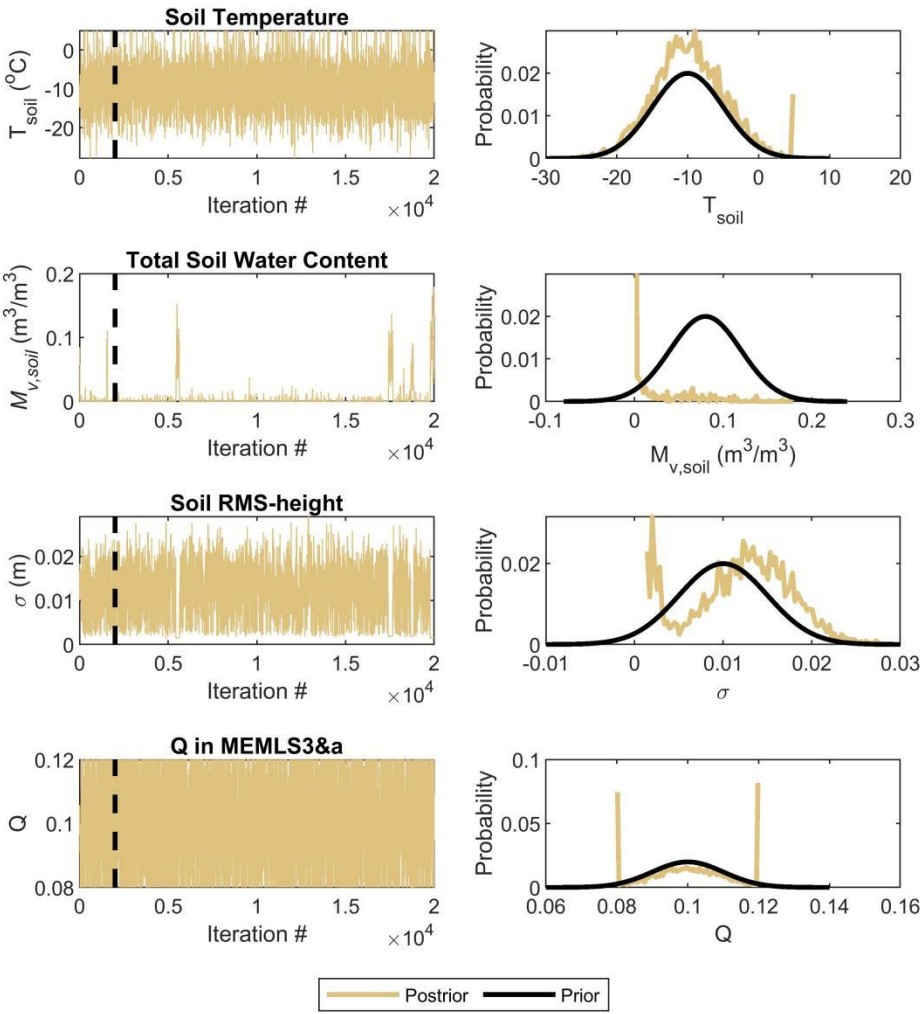

**Figure 6: MCMC chain of other soil and model variables (first column) and their posterior distributions compared with the prior distributions (second column) for Pit 49.**

### 4.2 Estimation of snow depth and snow water equivalent

Fig. 7 shows the MCMC-estimated snow depth (SD) (a) and SWE (b), from the averages of MCMC chains after the burn-in period. The BASE-AM algorithm corrects the underestimation of VIC priors: the original biases for SWE are -49.6, -16.8, -54.5 and -87.9 mm for IOP 1, 2, 3, 4, respectively, whereas the biases are changed to 12.3, 25.8, -15.3 and -34.8 mm, respectively, after retrieval. Fig. 8 summarizes the root-mean-squared error (RMSE). On average, the BASE-AM algorithm reduces the posterior RMSE to about half of the prior. The SWE RMSE for all snowpits is within 30 mm, whereas the SD RMSE for all snowpits is close to 10 cm. For IOP 2, the posterior SWE RMSE is higher than prior RMSE, because of an

overestimation of both SD and snow density. In IOP 4, we observed a strong influence of snow density accuracy on SWE
retrieval, when the SD estimation aligns with the fluctuation of SD caused by snowfall and snow compaction events.

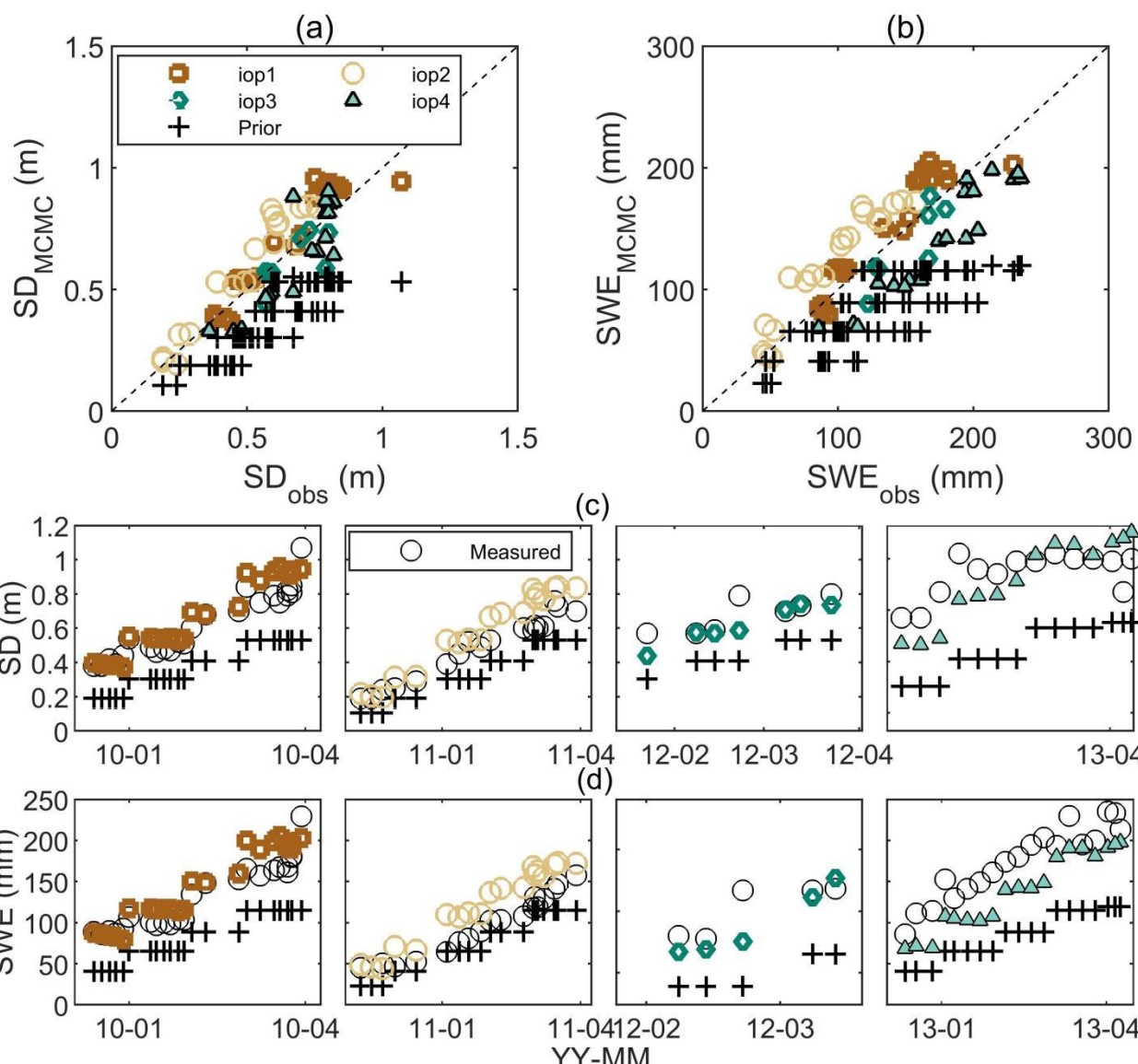

**Figure 7: BASE-AM estimated snow depth (SD) versus observed snow depth at Sodankylä (a, c), and BASE-AM estimated snow water equivalent (SWE) versus observed SWE (b, d). Scatterplots are shown in (a-b), whereas the time series are shown in (c-d).**

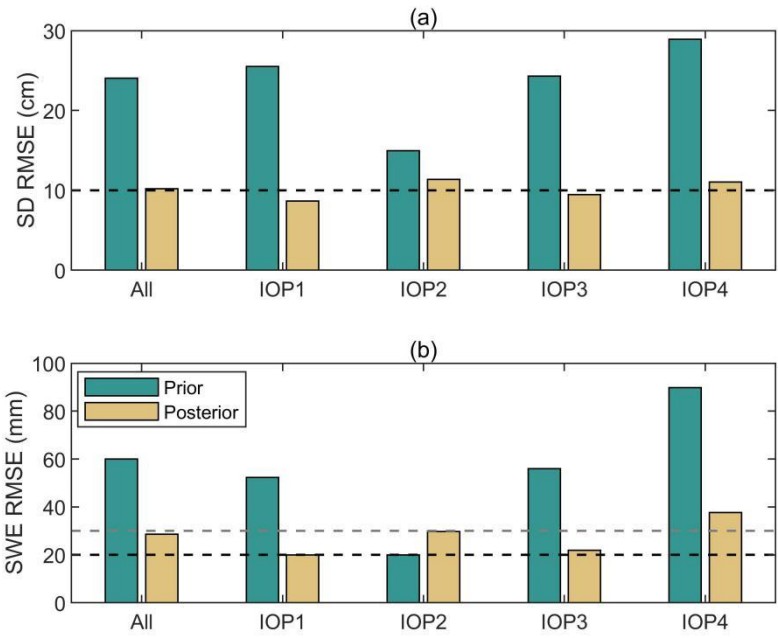

**Figure 8: Summary of root-mean-squared error (RMSE) for SD (a) and SWE (b) for different IOPs. The black dash lines are 10 cm RMSE for SD. The black and grey dash lines are 20 and 30 mm RMSE for SWE.**

### 4.3 Estimation of snow microstructure

Fig. 9 shows a comparison between the MCMC-estimated exponential correlation length ($p_{ex}$), $p_{ex}$ converted from the measured $D_g$ using the conversion equation in Pan et al.(2017), as $p_{ex}=0.227+0.126*\log(D_g)$, and a $p_{ex}$ fitted from the backscattering measurements using MEMLS3&a. The fitted $p_{ex}$ was calculated based on the snowpit and soil measurements, using an adjustable soil roughness ($\sigma$) to match the X-band and a scaling factor for $p_{ex}$ to match the other frequencies. The $p_{ex}$ scaler was constant along the snow profile but varies for different snowpits.

Fig. 9(a) shows that when $p_{ex}$ converted from measured $D_g$ is high in one IOP, the MCMC-estimated $p_{ex}$ is also high. However, Fig.9 (b) shows that within each IOP, the correlation between $p_{ex}$ converted from $D_g$ measurements and MCMC-estimated $p_{ex}$ is low. The low correlation results from uncertainties in $D_g$ observation, the $D_g$-$p_{ex}$ conversion equation, and probably uncertainties in backscattering observation and MEMLS3&a as well. Fig.9(c) shows if $p_{ex}$ is fitted from radar measurements using the same MEMLS3&a model, it matches significantly better with the MCMC-estimated $p_{ex}$. The result

in Fig. 9(c) indicates BASE-AM may be able to estimate snow microstructure parameter and SWE together, if the observation model can accurately describe the relationship between snow/soil parameters and radar measurements.

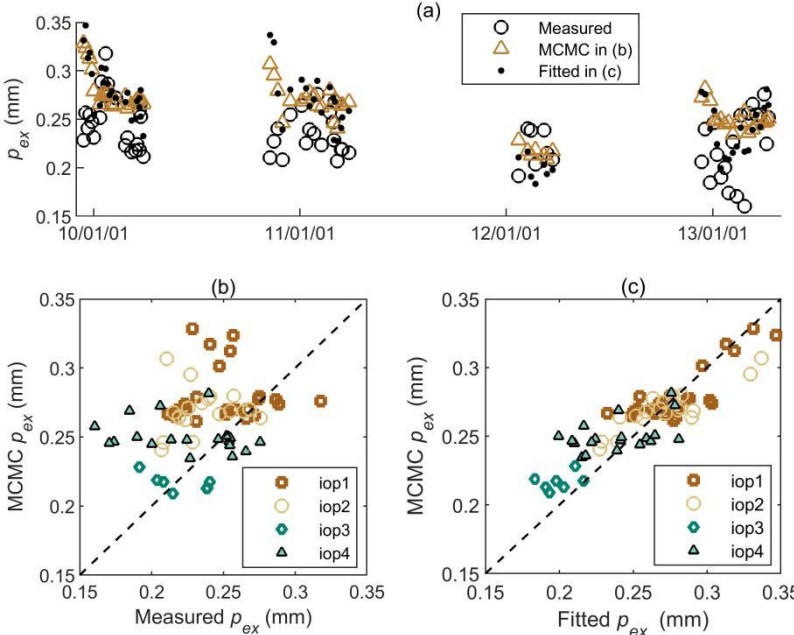

**Figure 9: Time series (a) and scatterplots of BASE-AM estimated profile-average exponential correlation length ($p_{ex}$) compared with $p_{ex}$ converted from measured $D_g$ from snowpits in (b), and with the fitted $p_{ex}$ in (c). The fitted $p_{ex}$ comes from the use of MRMLS3&a to match the measured backscattering coefficient at three frequencies.**

Fig. 10 uses 2-D distribution maps to show the relationship between layer thickness and exponential correlation length ($p_{ex}$) in the MCMC chains after the burn-in period, for Pit 49. After the burn-in period, all combinations of estimated variables can reproduce the observation. Fig. 10 shows that the measured backscattering coefficients specify an up-down flipped logarithm-like relationship between layer thickness and $p_{ex}$: when layer thickness is high, $p_{ex}$ is low, and vice versa; both parameters begin to saturate when the other approaches a high value. The area in dark purple is better represented in the MCMC chain, and thus has the highest probability, considering both observations and priors. This high probability area forms a logarithm-shape stripe instead of converging to a single point, indicating that estimating SWE using a robust priori information requires a global optimization instead of a local optimization. In Fig.10 (a) and (b), the observation changes the priors of layer thickness and $p_{ex}$ at the intersection of red dash lines to the posteriors at the intersection of black dash lines. However, in Fig. 10(c), there is more uncertainty in the estimation of the surface-layer $p_{ex,}$ than that of the bottom-layer, because the sensitivity of radar backscatter to volume scattering decreases with decreasing $p_{ex}$.

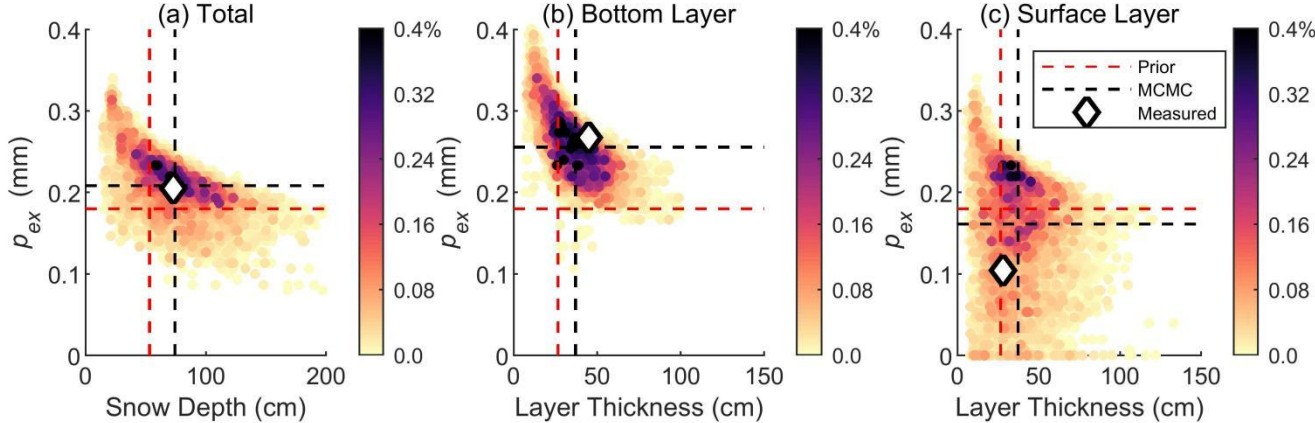

 **Figure 10: 2D histogram of probability between layer thickness and exponential correlation length ($p_{ex}$) from the MCMC chain after burn-in period for Pit49: (a) for the entire snowpit, where the snow depth and mass-weighted average $p_{ex}$ are presented, (b) for the bottom layer, (c) for the surface layer. The red dash lines represent the means of priors. The black dash lines represent the means of posteriors, which are the MCMC retrieval results. The white-face diamonds represent the measured SD (or layer thickness) and $p_{ex}$ converted from the measured $D_g$ from the snowpit measurements.**

## 5 Discussion

### 5.1 Concerning snow density, soil roughness and soil moisture

Backscattering coefficient at 10.2 GHz ($\sigma_0^{VV}{}_{10.2}$) is determined by snow density, soil liquid water content, and soil roughness, and we have shown the sensitivity of $\sigma_0^{VV}{}_{10.2}$ to the first two variables in Fig. 1. However, for each snowpit, a single observation of $\sigma_0^{VV}{}_{10.2}$ is insufficient to determine three variables together. From our retrieval result, we found that regardless of the measured profile-average snow density ranging from 150 to 300 kg/m$^3$, the BASE-AM algorithm consistently estimated a snow density of 215.5 kg/m$^3$, with a standard deviation of only 7.0 kg/m$^3$ across all snowpits. According to our simulations, the sensitivity of the microwave signal to snow density is lower than the soil parameters. This indicates that snow density is difficult to retrieve based on a single low-frequency microwave observation, unless soil liquid water content and soil roughness are provided (Gao et al., 2023), or more observations are provided (Lemmetyinen et al.,2016b).

As to the other two soil variables, the results in Section 4 are based on the default BASE-AM algorithm configuration to estimate total soil water content and soil roughness simultaneously. To further explore the algorithm, we conducted an additional experiment estimating only the soil moisture, using a fixed soil roughness of 1 mm. We found that when both the two soil parameters were estimated, the simulated backscattering coefficient in MCMC is closer to the observations, which is 0.43 dB RMSE compared with 0.52 dB; in addition, the accuracy of MCMC-estimated SD and SWE is also slightly higher, with RMSEs for all snowpits as 10.2 cm and 28.67 mm, respectively, compared with 10.71 cm and 30.14 mm, respectively. However, Fig. 11 shows, when the soil roughness is fixed, the temporal variation of estimated soil liquid water content matches better with the sensor measurements, which means the soil liquid water content becomes retrievable. This suggests a

possible strategy where the soil roughness is estimated at a single point early in the season, and the result used for the rest of the period if there is a desire to better estimate soil moisture dynamics from the radar data.

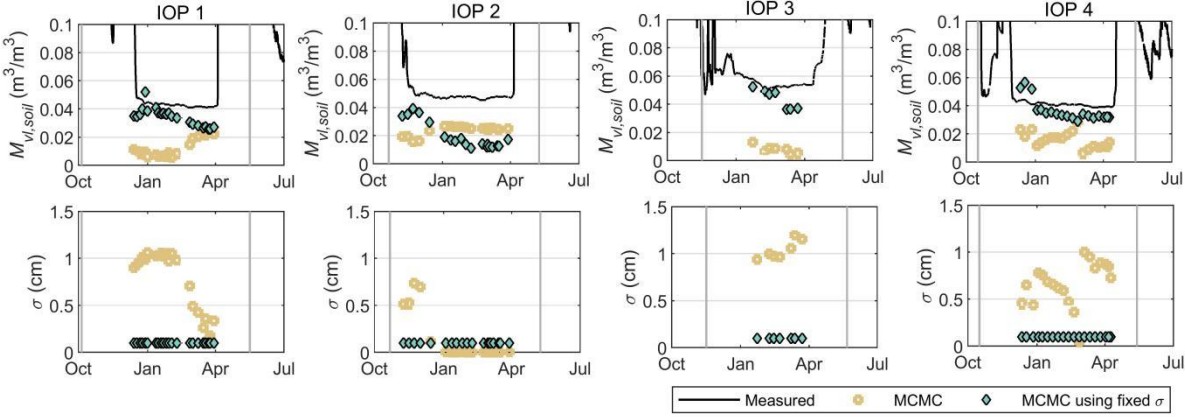

**Figure 11: Comparison of MCMC-estimated soil liquid water content ($m_{vl,soil}$) and soil roughness ($\sigma$) when both soil moisture and soil roughness are estimated (light brown circles) or only the soil moisture is estimated (green diamonds). The MCMC $m_{vl,soil}$ is calculated from the MCMC-estimated total soil water content and soil temperature using the same unfrozen soil content module in the forward model.**

**5.2 The influence of the number of modeled snow layers on the retrieval**

Fig. 12 shows the MCMC-estimated SD and SWE when the snow is assumed to have a single layer. The same snow and soil priors were used. When the one-layer snow assumption is used, BASE-AM cannot fully correct the underestimation of SWE prior. This occurred because the presence of a surface snow layer was overlooked. This layer generates very small volume scattering, rendering it nearly transparent to radar. Therefore, it is crucial to acknowledge the existence of such a surface

layer; otherwise, the total SD will be underestimated. At the same time, because the equivalent one-layer $p_{ex}$ represents that from the bottom layer, it will be larger than the profile-average $p_{ex}$ (from both the $D_g$ measurements and model fittings) (see Fig. 13). As summarized in Table 2, the snow retrieval result using two-layer assumption has a higher absolute bias and lower RMSE. The one-layer assumption underestimates SD and SWE and overestimates profile-average $p_{ex}$, especially for IOP 3 and IOP 4.

Fig. 14 makes a comparison of the SD-$p_{ex}$ relationship in MCMC chains determined by the same backscattering measurements for Pit49 using different snow layer assumptions. For the one-layer retrievals, the same three-frequency backscattering coefficients determined higher profile-average $p_{ex}$. Thus, the snow layer assumption can influence the SD and $p_{ex}$ estimation results, despite the priors utilized. At the same time, Fig. 14 agrees with the common sense that when more variables are estimated, the uncertainty of the variables increases.

Therefore, as a short summary, we recommend considering two layers since radar tends to overlook the surface snow layer of small grain size. Additionally, based on the balance between the current numbers of radar observations and predicted

variables, introducing more layers provided little or no improvements in SWE estimation, unless a reliable prior for snow stratigraphy detail becomes available.

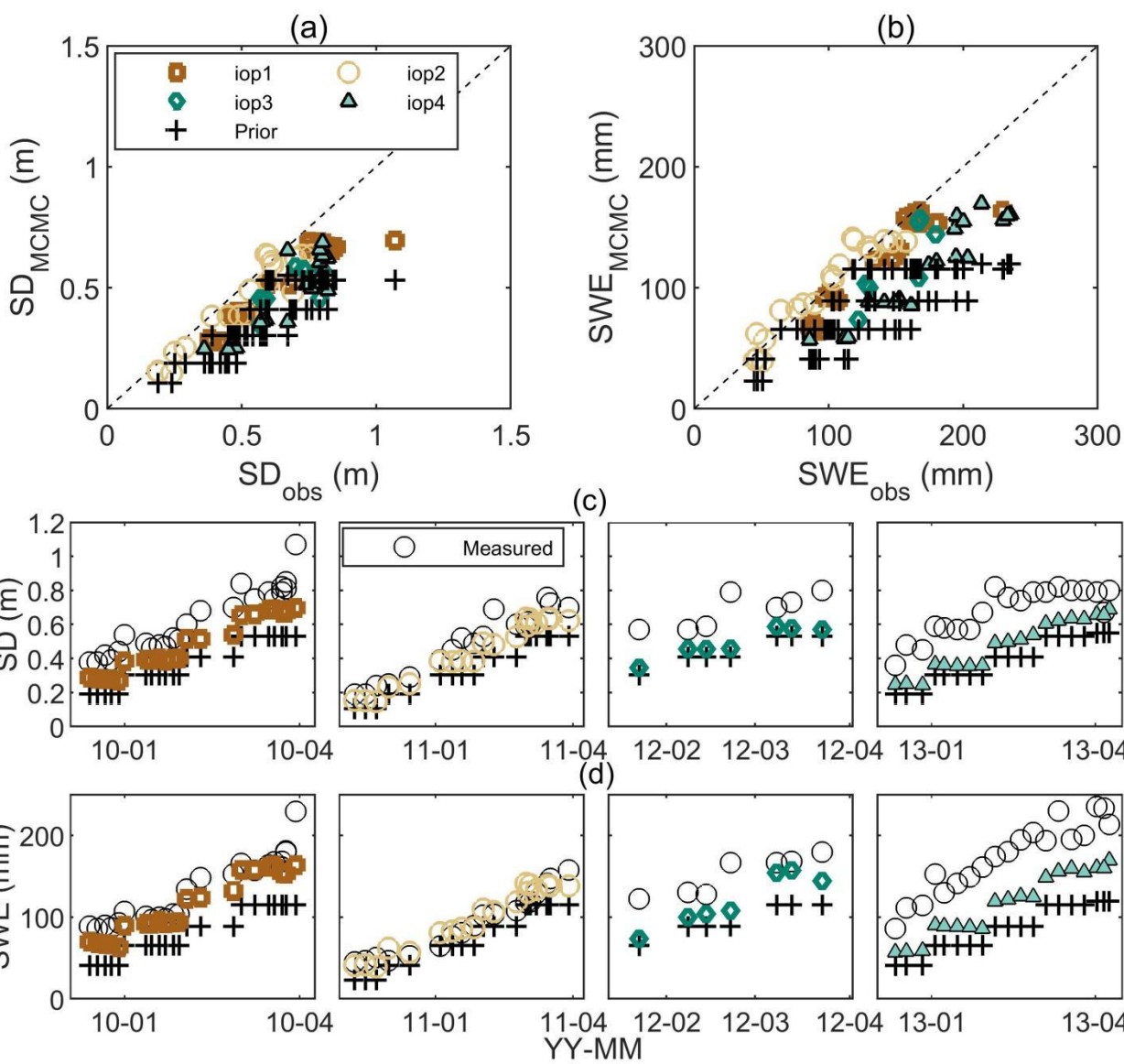

**Figure 12: BASE-AM estimated snow depth (SD) (a, c) and snow water equivalent (SWE) (b, d) using one-layer snow assumption compared with the measurements.**

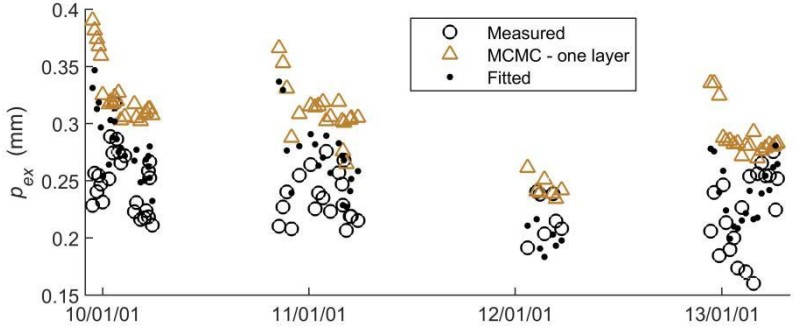

**Figure 13: BASE-AM estimated $p_{ex}$ using one-layer snow assumption compared with the measurements.**

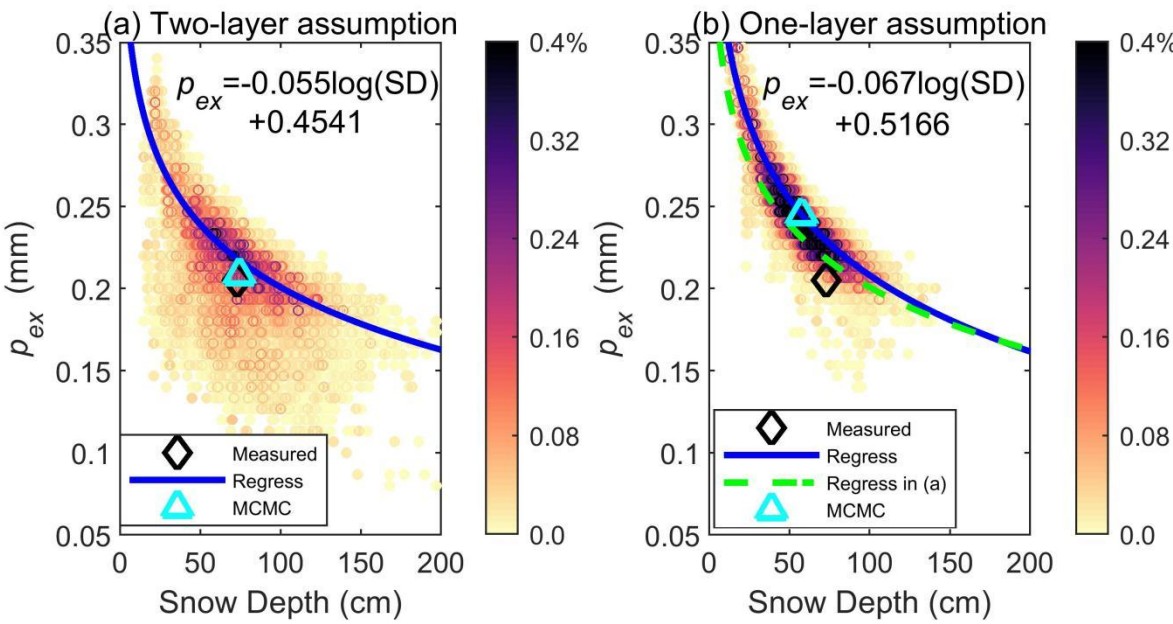

**Figure. 14: Comparison between the SD and profile-average $p_{ex}$ relationships in MCMC chains using two-layer (a) and one-layer (b) snow assumptions, respectively for Pit 49. The highest probability point at each snow depth was calculated and used to fit a logarithmic equation between SD and $p_{ex}$, and labeled as blue curve in both figures. The equation fitted in (a) are labeled as green dash curve in (b) to make a comparison. The measurements are labeled in black diamonds; the MCMC-estimated SD and $p_{ex}$ are labeled as blue triangles in both plots.**

**Table 2. Summary of SD, SWE and profile-average exponential correlation length ($p_{ex}$) estimation error using two-layer and one-layer snow assumptions**

| Parameters | Period | Mean bias | | | RMSE | | |
|---|---|---|---|---|---|---|---|
| | | Prior | Posterior using two-layer snow assumption | Posterior using one-layer snow assumption | Prior | Posterior using two-layer snow assumption | Posterior using one-layer snow assumption |
| SD (cm) | All | -22.3 | 2.4 | -13.4 | 24.0 | 10.2 | 15.9 |
| SWE (mm) | All | -51.6 | 0.25 | -23.5 | 60.0 | 28.7 | 35.5 |
| Profile-average $p_{ex}$ (mm)* | All | -0.078 | 0.004 | 0.045 | 0.086 | 0.019 | 0.0483 |

**\*Note: To compare with the MCMC $p_{ex}$, here we used the fitted $p_{ex}$ instead of the snowpit-measured $p_{ex}$ as the reference; the source of fitted $p_{ex}$ can be found in Section 4.3.**

## 6 Conclusions

In this paper, we developed a Bayesian-based Algorithm for SWE Estimation for Active Microwave (BASE-AM) to retrieve the snow and soil parameters from a site in Sodankylä, Finland based on X- and dual-Ku VV-pol. backscattering coefficients using biased SWE and uninformative snow microstructure priors. Results show that by predetermining the snowpack to have two layers, SD can be retrieved with a RMSE of 10.2 cm in 0-1 m range, and SWE can be retrieved with a RMSE of 28.7 mm in 0-200 mm range. The radar backscattering observations can correct the bias of SD prior from -22.3 cm to 2.4 cm using two-layer snow assumptions, but it can be only corrected to -13.4 cm if one-layer snow assumption is used. Results assuming a single layer are significantly less accurate, despite using the same priors.

By iteratively updating several snow and soil variables in the MCMC chain and comparing the prior and posterior distributions, we showed that the key variables required to be estimated in BASE-AM are layer thickness, layer microstructure parameter and soil liquid water content. The backscattering coefficients in snow and soil are not sensitive to temperature. Backscattering intensity at low frequency is sensitive to snow density, but density cannot be easily retrieved. The polarization splitting parameter ($Q$) in MEMLS3&a can be fixed unless cross-pol. backscattering observations are introduced.

Overall, our results indicate that active remote sensing observations coupled with a generic priori information and a two-layer retrieval scheme can support estimation of SWE. Moreover, it's essential to note that the prior setting is not a fixed component of the MCMC algorithm. For application in other regions, additional research may be necessary to reset the prior for $Q$ and the relative thickness between two layers. For instance, observations of depth hoar in tundra snow types indicate it occupies only 1/3 of the entire snow depth and can be adjusted accordingly (King et al., 2018; Saberi et al., 2021). Snow strategraphy from in-situ measurements or multiple-layer snow process model simulations (Pan et al., 2023) can also be incorporated to improve the retrieval performance. However, the example presented in this paper is currently computationally expensive, limiting its feasibility for generating products at a global scale. For global application, firstly, we can reduce the number of predicted variables. For instance, snow and soil temperatures can be replaced with predictions

from a land surface model. Soil roughness can be predetermined during snow-free periods, as the example in Gao et al.
(2023). For snow density and soil moisture, methods like those in Zhu et al. (2018) can be employed to avoid solving for both, or knowledge from one can be introduced to solve for the other (Kumawat et al., 2022; Gao et al., 2023). Ideally, the system would only need to iterate the layer thicknesses and layered snow microstructures, with the MCMC algorithm seeking global optimization instead of local optimization. Secondly, the length of chains can be reduced by monitoring the convergence of estimated variables as outlined in Section 7.3 of Pan et al.(2017). Thirdly, the computation of scattering
coefficients in MEMLS3&a based on IBA can be accelerated by using a machine learning model.

## Appendix. The soil dielectric constant model

The soil dielectric constant model utilized in this paper was developed from the Mironov et al. (2004) model and revised according to the soil dieletric constant measurement experiment conducted by Jinmei Pan using Agilent vector network analyzer at the Beijing Normal University in China. An introduction of the experiment can be found in Wu et al.(2022). Here
a brief introduction of this model is provided, with more details to be published. The measurements utilized to develop this model cover a wider soil texture than Wu et al.(2002), from loamy sand (77.27 % sand, 16.02 % silt, 6.71 % clay) to silty clay loam (17.49 % sand, 49.68 % silt, 32.82 % clay). The gravimetric soil water content of all samples varies from 3% to 60%, and the soil temperature varies from -30 to 20 °C. Soil dielectric constants were measured from 200 MHz to 20 GHz continuously at 200 MHz steps.

The real and imaginary part of the squared root of soil dielectric constants ($\varepsilon_{soil}$) are the refractive index (RI) ($n$) and the normalized attenuation coefficient (NAC) ($\kappa$).

$$n + i\kappa = \sqrt{\varepsilon_{soil}} = \varepsilon'_{soil} + i\varepsilon''_{soil} \tag{5}$$

$n$ and $\kappa$ of thaw soil are modeled as (Mironov et al., 2004):

$$n_s = \begin{cases} n_d + (n_b - 1)W, & W \leq W_B \\ n_d + (n_b - 1)W_B + (n_u - 1)(W - W_B), & W \geq W_B \end{cases} \tag{6}$$

$$\kappa_s = \begin{cases} \kappa_d + \kappa_b W, & W \leq W_B \\ \kappa_d + \kappa_b W_B + \kappa_u(W - W_B), & W \geq W_B \end{cases} \tag{7}$$

where, the subscripts of $n$ and $\kappa$, which include $d$, $b$, and $u$ represent dry soil solids, bound water, and free water, respectively. In the soil-water system, a fraction of bound water adheres to soil solids, which has a different dielectric property compared with free water. The maximum allowable bound water content is denoted as $W_B$. If the total water content ($W$) is lower than $W_B$, then the model will only contain two components, as the first lines of equation (6) and (7); otherwise, it will contain three components.

$n$ and $\kappa$ of frozen soil are modeled as (Mironov et al., 2017):

$$n_s = \begin{cases} n_d + (n_b - 1)W, & W \leq W_B \\ n_d + (n_b - 1)W_B + (n_t - 1)(W - W_B), & W_B \leq W \leq W_U \\ n_d + (n_b - 1)W_B + (n_t - 1)(W_U - W_B) + (n_i - 1)(W - W_U), & W \geq W_U \end{cases} \tag{8}$$

$$\kappa_s = \begin{cases} \kappa_d + \kappa_b W, & W \leq W_B \\ \kappa_d + \kappa_b W_B + \kappa_t(W - W_B), & W_B \leq W \leq W_U \\ \kappa_d + \kappa_b W_B + \kappa_t(W_U - W_B) + \kappa_i(W - W_U), & W \geq W_U \end{cases} \tag{9}$$

where, the subscripts of $n$ and $\kappa$, which include $d$, $b$, $t$ and $i$ represent dry soil solids, bound water, transient water, and ice, respectively. Soil solids and bound water have the same physical meanings for frozen soil and for thaw soil. When soil temperature decreases from above-zero °C to sub-zero °C, not all water will immediately freeze into ice. The total volumetric fraction of unfrozen water is called the unfrozen water content ($W_U$), which can be calculated as a function of temperature and clay fraction. We followed the setting in Mironov et al.(2017) that the bound water will not freeze, and the unfrozen soil water that exceeds the bound water is considered as the transient water.

The key of soil dielectric constant model is to model $W_B$, $W_U$, and $n$ and $\kappa$ for all components. We utilized the same $W_B$ and the same $n$ and $\kappa$ for soil solids as a function of clay fraction in Mironov et al. (2004). $n$ and $\kappa$ of ice refer to the temperature-dependent equation utilized in MEMLS3&a.

The dieletric constants of water components can be modeled by Debye's equation as:

$$\varepsilon = \varepsilon_\infty + \frac{\varepsilon_0 - \varepsilon_\infty}{1 - i2\pi f\tau} + i\frac{\sigma}{2\pi f\varepsilon_r} \tag{10}$$

where, $\varepsilon_\infty$ is the dielectric constant in the high-frequency limit, $\varepsilon_0$ is the static dielectric constant in low-frequency limit, $f$ is frequency (Hz), $\tau$ is the relaxation time (s), $\sigma$ is the effective conductivity (S/m), and $\varepsilon_r$ is the dielectric constant for free space ($8.854\times10^{-12}$ F/m). Later, $\varepsilon$ can be transferred to $n$ and $\kappa$ using equation (5).

For each component, it requires to determine $\varepsilon_0$, $\varepsilon_\infty$, $\tau$, and $\sigma$. We adapted some existing equations from Mironov et al. (2004) and Stogryn (1971), and fitted the remaining according to measurements. We utilized the samples with $W<W_B$ to fit temperature-dependent bound water parameters (see Table 3), and then iteratively fit $W_u$ and transient water parameters using the MCMC approach. Table 3 lists the details of water component models, and $W_U$ was determined as:

$$W_U = min(A \times |T|^B, W) \tag{11}$$

$$A = a_c C + a_{mv} W$$

$$B = b_c C + b_{mv} W$$

where, $C$ is soil clay content (%), $T$ is soil temperature (°C), $a_c$=0.00306, $a_{mv}$=0.394, $b_c$=0.00582, and $b_{mv}$=−1.073.

Table 3. Sources or equations of $\varepsilon_0$, $\varepsilon_\infty$, $\tau$, and $\sigma$ for different soil water components

| | $\varepsilon_0$ | $\varepsilon_\infty$ | $\tau$ | $\sigma$ |
|---|---|---|---|---|
| Free water | Stogryn (1971) as a function of $T$ | | | Mironov et al. (2004) as a function of $C$ |
| Bound water | $24.9 + 0.0685T$ | $80.8 + 0.715T$ | $26.04\times10^{-12}$ | Mironov et al. (2004) as a function of $C$ |
| Transient water | 4.78 | 77.90 | $24.0\times10^{-12}$ | 0.3913 |

\* $T$ and $C$ are soil temperature (°C) and soil clay content (%)

Fig. 15 shows an example of the model utilized to predict the measured soil permittivity (real part of dielectric constants) at

100 MHz by the Decagon 5 TM sensor in Sodankylä. The simulation utilized the measured liquid water content and

measured soil texture. The total soil water content comes from the measured liquid water content before freezing. To

calculate soil permittivity, we have used a fully-independent model compared to from the Decagon 5 TM sensor. Fig. 12(a)

shows our model is capable of predicting the change of soil permittivity with soil temperature. Fig.12(b) shows the simulated

soil permittivity is highly consistent with the observations at 10 cm to 80 cm depth below the soil surface. The mean bias is

0.0838, with a RMSE of 0.0569. In addition, the model overestimates the measured permittivity for the single top layer at 5

cm, which is influenced by air above soil and organic matter. Including the 5-cm layer, the mean bias is 0.129 with a RMSE

of 0.198. It indicates that the soil dielectric constant model described here is suitable to be used as part of the forward model

in our paper.

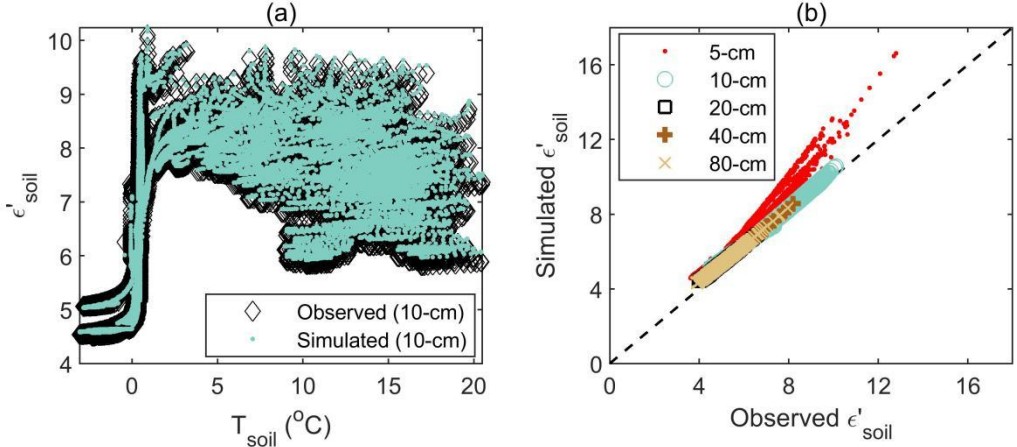

**Figure 15: Simulated and observed real part of soil dielectric constants ($\varepsilon'_{soil}$) by Decagon 5 TM sensor at 100 MHz: (a) sensitivity to temperature for soil measured at 10 cm; (b) scatterplots for soil measured at different depths.**

**Data availability:** The NoSREx datasets are available after registration on the ESA Earth Observations Campaign Data

portal (https://earth.esa.int/web/guest/campaigns).

**Author contribution**: JL provided the NoSREx datasets, and preliminary analysis of these datasets was conducted by JP, MD, and JL prior to retrieval. MD and DL provided the MCMC algorithm ideas and tools, which were later implemented and revised by JP to conduct the SWE retrieval experiments. All co-authors participated in the analysis of the MCMC results

and collaborated on writing and revising the manuscript together.

**Competing interests**: The authors declare that they have no conflict of interest.

## Acknowledgment

The authors want to thank the NoSREx team for their hard work and dedicated efforts for completing this snow experiment and providing the great dataset to support our study. We want to thank Kimmo Rautiainen for providing the soil frost depth measurement for us to do more analysis, and thank Simon Yueh and Richard Kelly for providing valuable comments when we were studying the first MCMC outputs. We dedicate this study to the memory of Dr. Joshua King, who tragically passed away February 21, 2023. Josh's pioneering measurements and keen insight into estimation of snow water equivalent from

radar observations were an inspiration for us and for the entire community, and he will be deeply missed. This work was funded by the National Natural Science Foundation of China (Grant No. 42090014), the National Key Research and Development Program of China (Grant No. 2021YFB3900104), NASA grant 80NSSC17K0200, and was financially supported by the China Scholarship Council and the OSU (Ohio State University) Presidential Fellowship. The three reviewers are acknowledged for their helpful comments with which greatly improved the manuscript.

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
