# Peer review of "Snow water equivalent retrieved from X- and dual Ku-band scatterometer measurements at Sodankylä using the Markov Chain Monte Carlo method"

_The Cryosphere, 2023_

## Author Response (AR1)

20 Nov 2023

**Editor decision: Reconsider after major revisions (further review by editor and referees)**

by Homa Kheyrollah Pour

**Public justification (visible to the public if the article is accepted and published)**:

Thank you for submitting your manuscript to TC Journal. We have received feedback from three reviewers, each of whom has raised significant concerns. In light of this, we recommend that your manuscript undergo major revisions before reconsideration. As you prepare your revised submission, please thoroughly address each point raised by the reviewers. Along with your revised manuscript, kindly provide a detailed list of responses to the reviewers' comments, demonstrating how you have addressed their concerns

Dear editor, thank you so much for handling our paper. We have carefully read questions and suggestions from the three reviewers and made corresponding improvements in our revised manuscript. Here attached our responses to the reviewers' comments. Thank you again.

**Anonymous Referee #1, 04 Oct 2023**

Radar snow sensing is challenging due to the intricate interactions between microwave and snowpack, which is defined by a variety of parameters. The paper makes a good attempt at addressing this topic using the MCMC approach and MEMLS3&a model, and suggests that accurate prior information about snow microstructure is not necessary for SWE retrieval. Overall, the approach is novel and interesting. However, there are a few issues to be clarified before I can recommend it for publication:

We really appreciate your hard work and insightful comments in evaluating our paper. Please allow us to reply to your questions point by point:

(1) The study relied on the MEMLS3&a model for forward simulations. Both the study and the MEMLS3&a model were validated using the same data set from NoSREx. The authors would show a more robust validation if the retrieval approach could be tested in another region with different snow properties than those seen in the NoSREx data set. In addition, please specify the uncertainty and applicability of MEMLS3&a model.

Reply: The utilization of a forward model that accurately describes observed signals with acceptable errors forms the foundation for its application in retrieval studies. While not presented in this paper, we conducted a successful forward simulation test at Sodankylä before performing the retrieval process, which demonstrated that no calibration of internal equations was necessary for MEML3&a.

MEMLS3&a is an adaptation of the physically-based MEMLS3 model for passive microwave applications, which has been validated in various studies to exhibit an error range of 5-15 K. The snow microstructure used for active may need to be tuned larger compared to passive; however, the two correlation lengths are highly correlated (Lemmetyinen et al., 2018). Therefore, it will not influence the SWE retrieval based on active measurements alone.

The primary uncertainty associated with MEMLS3&a arises from the empirical parameter, Q, which quantifies the extent to which scattered wave is polarized cross-polarization rather than co-polarization. This makes MEMLS3&a actually more adaptive to different regions and snow properties, but also introduces an additional parameter to be estimated. Therefore, if this method is extended to another region, we would suggest considering two options:

Option 1: Conduct research to calculate Q based on co- and cross-polarization measurements, and determine whether the original prior for Q needs to be revised.

Option 2: Incorporate cross-polarization observations into the Markov Chain Monte Carlo (MCMC) algorithm, enabling direct prediction of Q by MCMC.

Reference:
Lemmetyinen, J., Derksen, C., Rott, H., Macelloni, G., King, J., Schneebeli, M., Wiesmann, A., Leppänen, L., Kontu, A., and Pulliainen, J.: Retrieval of effective correlation length and snow water equivalent from radar and passive microwave measurements, Remote Sens., 10, https://doi.org/10.3390/rs10020170, 2018.

(2) I also wonder why only VV polarization was used while HH and VH measurements could provide more information of snow.

Reply: Since MEMLS3&a physically calculates the total scattering using IBA while empirically determining cross- and co-polarizations, we didn't use all polarizations. However, as we relied in (1), VH measurement is currently involved in setting the prior for Q, and in future it can be directly utilized as observations in MCMC.

We agree that if a more physically-based model can accurately predict the VH/VV ratio without introducing any additional parameters, it could lead to further improvement in the estimation of SWE.

(3) For the retrieval approach, why using two layers instead of multiple layers? Was part of the prior-knowledge provided by defining a depth hoar layer?

Reply: This paper is an extended study of Pan et al.(2017), in which two layer is found to be the minimum number of layers that can practically simulate a similar response of $T_B$ to SWE as multiple layers. The primary objective of this study was to assess the feasibility of retrieval with minimal prior knowledge, therefore we started with a two-layer scheme instead of a more complex multiple-layer scheme.

The part of prior-knowledge can be considered from a depth hoar layer, or simply an older snow layer compared to new snow of snow grain size. We didn't use in-situ measurements to determine the prior for relative thicknesses of two layers, but the users can do so according to their research interests. It's important to note that the prior knowledge setting is not a fixed part of Markov Chain Monte Carlo.

Reference:
Pan, J., Durand, M. T., Vander Jagt, B. J., and Liu, D.: Application of a Markov Chain Monte Carlo algorithm for snow water equivalent retrieval from passive microwave measurements, Remote Sens. Environ., 192, 150–165, https://doi.org/10.1016/j.rse.2017.02.006, 2017.

(4) Could the authors specify the penetration depth of each frequency and their roles/contributions in estimating SWE and other parameters?

Reply: Penetration depth is sensitive to snow grain size and varies with details in snow stratigraphy. Here we can only provide an example. According to our simulation by calculating the two-way transmissivity of snow layer ($t_0^2$ in Wiesmann& Matzler, 1999) until it reaches 1/e, if $p_{ex}$ is 0.18 mm, the penetration depth of 10.2, 13.3 and 16.7 GHz are 17 m, 9.1 m and 5.1 m, respectively. If pex increases to 0.35 mm, the penetration depths decrease to 8.9 m, 4.1 m and 2.1 m respectively. The other input variables to support this calculation are: snow T=-5 degC, snow density=200 kg/m$^3$, incidence angle=50 deg.

10.2 GHz is sensitive to surface scattering instead of volume scattering, and thus provides information to solve soil permittivity and snow density. 13.3 and 16.7 GHz are sensitive to volume scattering and have different scattering coefficients at different frequencies, and thus they provide information to solve SWE and snow grain size.

As pointed out by the reviewer, we decide to revise line 183 from:

"This was done because we found that the penetration ability of radar at these frequencies have largely reduced the sensitivity to volume scattering from the surface snow layers of small grain sizes."
to:
"This was done because we found that the radar at these frequencies has a small sensitivity to volume scattering from the surface snow layers of small grain sizes."

References:
Wiesmann, A., Mätzler, C., 1999. Microwave emission model of layered snowpacks. Remote Sens. Environ. 70, 307–316. https://doi.org/10.1016/S0034-4257(99)00046-2.

(5) Table 1: The mean value of the pre-defined soil RMS-height is 1 mm or 1 cm?
Reply: It is 1 cm. Thank you for pointing this out.

(6) Section 3.3: how did you get the soil texture information?
Reply: We have added:
The soil texture was set as 70% sand, 29% silt and 1% clay, according to the measurements from NoSREx (Lemmetyinen et al., 2016).

(7) Fig.4 and 5: please explain about the high probability observed at the low and high ends of the parameter range (e.g. Fig. 5 row 4, column 2)
Reply: It means the these parameters were restricted within the permissible range defined in Table 1 by the MCMC algorithm. At the start of each iteration, a jump function is applied to the parameter value from the previous iteration, and for instance, if the snow temperature surpasses 0 degree Celsius, it will be mandatorily adjusted to 0 degree Celsius. Subsequently, the likelihood ratio calculation continues.

(8) Could you discuss about the options of different prior distributions and the potential impacts on the retrievals? How can readers select an optimum prior distribution?
Reply: In this paper, we applied robust priors for snow and soil parameters. As demonstrated in Pan et al.(2017), more accurate priors generally lead to better retrieval results. However, the retrieval performance will be degraded when employing a specific but strongly biased prior. Therefore, we recommend readers to begin with a set of general priors, as used in this paper, and they can refine them using more detailed and reliable information if available. The optimal prior, aligning with the truth, is theoretically unattainable.
Regarding the choice between lognormal and normal distributions, the advantage of a normal distribution lies in its mean value overlapping with the peak probability. This results in a posterior distributions with a similar shape, making them easier to comprehend.

**Comment on tc-2023-85', Anonymous Referee 2**

In this study, the BASE-AM method is proposed to retrieve SWE and snow depth with NoSREx data with uniform a priori snow microstructure. The study is an extended one of Pan et al., 2017. I have several questions about the applied method which needs to be solved before publication.

We really appreciate your time in evaluating our paper and sending insightful comments for further improvements. Our replies to your questions are listed as follows point by point:

Major comments:

1) In the introduction, Line 89-93, I do not understand this paragraph. The study has been completed. So conclusions should be given clearly here, Not as those assumptions.
Reply: Line 89-93 does not express an assumption; instead, it outlines what will be learned in the event of a successful retrieval and what will be gleaned if it is unsuccessful. Since the results have not been presented in the introduction, making this statement is appropriate. It is not an assumption but rather delineates what will be learned if the retrieval is successful.
To better set the tone in this paragraph, we will add, "We hypothesize, based on previous results for passive microwave (Pan et al., 2017) at this site, that the radar retrieval algorithm will also successfully estimate SWE using the same generic prior."

2) In the introduction, authors should discuss other studies apply correlation length to do radar SWE retrieval. For example, Lemmetyinen, et tal, 2018.
Lemmetyinen, J., Derksen, C., Rott, H., Macelloni, G., King, J., Schneebeli, M., ... & Pulliainen, J. (2018). Retrieval of effective correlation length and snow water equivalent from radar and passive microwave measurements. *Remote Sensing*, *10*(2), 170.
Reply: We have read this paper. In Line 68, we have added:
"Lemmetyinen et al. (2018) demonstrated that radar SWE retrieval can be supported by scaling the effective correlation length obtained from passive microwave observations, which, however, was fitted using snow depth measurements, too. "

3) Throughout the paper, it is not correct to use "priori". "a priori" is one word means knowledges known in advance, e.g. a priori SWE, a priori information. Make sure to revise them in this paper.
Reply: Prior is a technical definition in Bayesian estimation (Gelman et al., 1995), where it also means knowledges known in advance, but can be a probability distribution instead of a fixed value.

4) In Section 3.1, Line 183-188, I have concerns with these assumptions. Authors should apply in situ measurements to demonstrate these assumptions are reasonable. Also, do these assumptions work for all snow types? As far as I know, the assumptions do not work for tundra snow. In King et al. 2018, the surface slab layer has a larger density than that of the depth hoar layer. Also, 67% of SWE from the surface layer with thickness could be several times of that of the depth hoar layer.

King, J., Derksen, C., Toose, P., Langlois, A., Larsen, C., Lemmetyinen, J., ... & Sturm, M. (2018). The influence of snow microstructure on dual-frequency radar measurements in a tundra environment. Remote sensing of environment, 215, 242-254.

Reply: These assumptions are supported by in-situ measurements at Sodankyla, following a taiga snow type rather than tundra. For tundra snow, it is possible that a revised prior would be more effective. We have an example utilizing a similar prior to your description:

Saberi, N., Kelly, R., Pan, J., Durand, M., Goh, J., and Scott, K. A.: The Use of a Monte Carlo Markov Chain Method for Snow-Depth Retrievals: A Case Study Based on Airborne Microwave Observations and Emission Modeling Experiments of Tundra Snow, IEEE Trans. Geosci. Remote Sens., 59, 1876–1889, https://doi.org/10.1109/TGRS.2020.3004594, 2021.

To clarify, we will add "They (these constraints) were set according to a taiga snow type, simply for producing a more reasonable profile feature for these two parameters not very sensitive to backscattering observations. Therefore, these constraints can be revised or deleted if detailed prior knowledge is provided."

5) I have concerns about the retrieval efficiency. How much time does it need to get one retrieval point?  Line 190-191, such a large number of iterations makes me concerned about the method efficiency. Considering global retrieval, the amount of data could be extremely large.

Reply: Yes. Efficiency is a problem for the current algorithm. It used ten minutes for a single retrieval point. However, estimating all parameters, as done in this paper, may be unnecessary.

In the conclusion, we have added:

"For global application, firstly, we can reduce the number of predicted variables. For instance, snow and soil temperatures can be replaced with predictions from a land surface model. Soil roughness can be predetermined during snow-free periods, as the example in Gao et al. (2023). For snow density and soil moisture, methods like those in Zhu et al. (2018) can be employed to avoid solving for both, or knowledge from one can be introduced to solve for the other (Kumawat et al., 2022; Gao et al., 2023). Ideally, the system would only need to iterate the layer thicknesses and layered snow microstructures, with the MCMC algorithm seeking global optimization instead of local optimization. Secondly, the length of chains can be reduced by monitoring the convergence of estimated variables as outlined in Section 7.3 of Pan et al.(2017). Thirdly, the computation of scattering coefficients in MEMLS3&a based on IBA can be accelerated by using a machine learning model."

6) In the retrieval model, parameters should be physical. Also, measurement data should be discussed to show the correctness of these parameters. In Table 1, for correlation length, 5 mm correlation length is about >30 mm grain size. It is not possible. Also for rms height, 10 cm is not possible. 50% soil moisture means water polls.

Reply: In Table 1, 5 mm for correlation length, 10 cm rms height and 100% soil moisture are the maximum allowed values for parameters. These upper limits have minimal impact on the retrieval result unless set unrealistically small. For the actual parameter range selected by MCMC, it is suggested to refer to Figures 5 and 6 (4 and 5 in the old manuscript).

7) Note that SWE and snow depth have been applied as a priori information. In Zhu et al, 2018, the study shows that backscatter is a function of SWE and snow microstructure (Pex). How the results would be if uniform a priori SWE and snow depth are applied in retrieval. For example, 100 mm SWE and 0.5 m snow depth.

Reply: In the study by Zhu et al. (2018), a prior for scattering albedo, which is essentially a snow microstructure prior, was employed. In our work, we transitioned from a requirement for a snow microstructure prior to a biased SWE prior, as the latter is typically easier to obtain, for example, from land surface simulations. The prior used in our paper has been quite general, operating on a monthly scale with a standard deviation (uncertainty) equal to 50% of SWE. Since we can at least derive a SWE prior by summing up precipitation with Tair lower than 0°C, we feel it is unnecessary to complicate this retrieval problem further.

8) In Figure 9, note the probability is still focused near 0.2. I am curious what retrieval results will be obtained if we consider some extreme a priori Pex. For example, Pex = 0.05 mm or Pex = 0.4 mm. If results are still acceptable, it can be better to conclude that no precise Pex needed for retrieval.

Reply: For the case in Figure 9 (now Figure 10), the posterior pex is near 0.2 mm. However, we got a maximum pex estimation of 0.35 mm for other cases, presented in Figure 8 (now Figure 9).

Our choice of a pex prior centered at 0.18 mm is based on the generic prior from our previous studies, as opposed to a 'local' prior informed by in-situ observations.

The sensitivity tests could be interesting, especially if we only vary the prior mean while keep the pex uncertainty constant. However, this sensitivity test is outside scope of the paper. We have added a note in Section 3.1: "The prior for snow exponential correlation length was uninformative, following N(0.18 mm, 0.09 mm) from Pan et al.(2017). It could be interesting to check the performance using a different uninformative pex prior mean, although it is beyond the scope of this manuscript.

Minor comments:

Line 81, It is not accurate to use the term "depth hoar layer" which does not exist for all types of snow. I think it may be better to use the surface layer and a bottom layer or the small grain size layer or large grain size layer.

Reply: Agree. It has been revised from "depth hoar layer" to "bottom layer" as suggested.

Line 83, does the study applied same a priori correlation length to both layers? it should be clearly pointed out.

Reply: We have added "... which follows a normal distribution as N(0.18 mm, 0.09 mm) for both snow layers".

In Figure 1, since data are discrete points, it is better to use markers rather than curves.

Reply: We have made revisions accordingly.

Equation (1), there is a typo. "P(M(x2))" should be "P_obs(M(x2))"

Reply: We have made revisions accordingly.

Figure 4: which layer is layer 2 (surface or bottom)? Need to point out clearly in caption.
Reply: We have added: "Lyr1 is for the bottom layer, whereas Lyr2 is for the surface layer."

**General Comments + Summary**

   This manuscript presents a Monte Carlo based method for estimating snow and land surface variables used to accurately recreate the observe backscatter in X- and Ku-bands as observed at a field site. The authors use two frameworks (single layer and two-layer) and suggest that the two-layer framework is more effective and can accurately estimate snow depth and SWE even if prior estimates of snow states are inaccurate. The paper is well-presented and well written, and the value of the proposed approach is generally clear (opportunity to better estimate SWE using remote sensing w/o accurate existing information). The work also has relevance understanding the capabilities (& limitations) of radar systems for estimating snow properties. With some minor revisions, the work appears ready for publication.

We really appreciate your hard work and insightful comments in evaluating our paper. Please allow us to reply to your questions point by point:

**General/Significant Comments**

- The authors mention the burn-in period and the ability of MCMC to 'stably reproduce' the observed backscatter signals. But based on the presented figures it is not clear why so many iterations are used (& if they are necessary) – the charts (e.g., Figure 3) appear to show what just looks like normally distributed noise – except for the zoomed one which shows some correction occurring in early iterations. It would be useful for the authors to include more information as to why the number of iterations was chosen and how many are minimally required to produce the same result.

    Reply: Thank you for pointing out this important issue. In this paper, we followed the same number of iterations as in Pan et al.(2017). However, it may be reduced, according to different setting of estimated variables and priors in future. We have added the following words at the end of this paper, "The length of chains can be reduced by monitoring the convergence of estimated variables as outlined in Section 7.3 of Pan et al.(2017). "

    It should be noted that in MCMC, a certain number of iterations are needed to achieve a global optimization instead of a local optimization. The problem in the snow microwave retrieval was there are so many combinations of variables that all can simulate a signal close to the observations. Especially, this problem is more serious in our paper because we estimated some insensitive variables, for example, snow temperature, soil temperature and Q. Therefore, the convergence needs to be checked for those of the estimated variables, not only that of the simulated backscattering. However, indeed the length of chains can be shortened if we only estimate layered snow thicknesses and microstructures.

- Similarly, there is limited information on the broader applicability of this approach. One which seems quite computationally expensive. Currently, there is no discussion to this end nor into if such an approach is feasible to implement using remotely sensed observations at a global scale. Is the approach expected to perform well in other conditions

(i.e., snow types, depths etc.)? I think if this is part of the objective, to be valuable beyond site specific studies/ground-mounted radar systems, some discussion is warranted.

Reply: We have added a new paragraph at the end of our paper:

"Overall, our results indicate that active remote sensing observations coupled with a generic priori information and a two-layer retrieval scheme can support estimation of SWE. Moreover, it's essential to note that the prior setting is not a fixed component of the MCMC algorithm. For application in other regions, additional research may be necessary to reset the prior for Q and the relative thickness between two layers. For instance, observations of depth hoar in tundra snow types indicate it occupies only 1/3 of the entire snow depth and can be adjusted accordingly (King et al., 2018; Saberi et al., 2021). Snow stratigraphy from in-situ measurements or multiple-layer snow process model simulations (Pan et al., 2023) can also be incorporated to improve the retrieval performance. However, the example presented in this paper is currently computationally expensive, limiting its feasibility for generating products at a global scale. For global application, firstly, we can reduce the number of predicted variables. For instance, snow and soil temperatures can be replaced with predictions from a land surface model. Soil roughness can be predetermined during snow-free periods, as the example in Gao et al. (2023). For snow density and soil moisture, methods like those in Zhu et al. (2018) can be employed to avoid solving for both, or knowledge from one can be introduced to solve for the other (Kumawat et al., 2022; Gao et al., 2023). Ideally, the system would only need to iterate the layer thicknesses and layer microstructures, with the MCMC algorithm seeking global optimization instead of local optimization. Secondly, the length of chains can be reduced by monitoring the convergence of estimated variables as outlined in Section 7.3 of Pan et al.(2017). Thirdly, the computation of scattering coefficients in MEMLS3&a based on IBA can be accelerated by using a machine learning model."

References:

Gao, X., Pan, J., Peng, Z., Zhao, T., Bai, Y., Yang, J., Jiang, L., Shi, J., and Husi, L.: Snow Density Retrieval in Quebec Using Space-Borne SMOS Observations, Remote Sens., 15, 1 – 19, https://doi.org/10.3390/rs15082065, 2023.

Kumawat, D., Olyaei, M., Gao, L., and Ebtehaj, A.: Passive Microwave Retrieval of Soil Moisture Below Snowpack at L-Band Using SMAP Observations, IEEE Trans. Geosci. Remote Sens., 60, 1 – 16, https://doi.org/10.1109/TGRS.2022.3216324, 2022.

King, J., Derksen, C., Toose, P., Langlois, A., Larsen, C., Lemmetyinen, J., Marsh, P., Montpetit, B., Roy, A., Rutter, N., and Sturm, M.: The influence of snow microstructure on dual-frequency radar measurements in a tundra environment, Remote Sens. Environ., 215, 242 — 254, https://doi.org/10.1016/j.rse.2018.05.028, 2018.

Saberi, N., Kelly, R., Pan, J., Durand, M., Goh, J., and Scott, K. A.: The Use of a Monte Carlo Markov Chain Method for Snow-Depth Retrievals: A Case Study Based on Airborne Microwave Observations and Emission Modeling Experiments of Tundra Snow, IEEE Trans. Geosci. Remote Sens., 59, 1876 – 1889, https://doi.org/10.1109/TGRS.2020.3004594, 2021.

Pan, J., Yang, J., Jiang, L., Xiong, C., Pan, F., Gao, X., Shi, J., and Chang, S.: Combination of Snow Process Model Priors and Site Representativeness Evaluation to Improve the Global Snow Depth Retrieval Based on Passive Microwaves, IEEE Trans. Geosci. Remote Sens., 61, 1 – 20, https://doi.org/10.1109/TGRS.2023.3276651, 2023.

- Can the authors add a workflow figure/graphic to show the modeling workflow. This would be useful to fully understand or replicate the approach, since there are several moving parts (i.e., the MCMC model, MW emission modeling, soils dielectric modeling, ground observations etc.)

Reply: Agree. We have added a flowchart in Figure 3.

- The paper does not include any time-series analysis of modeled SWE/depth vs. the observations. This may be out of scope of the paper, but I would imagine the accuracy of this approach is seasonally varied. In my opinion, it would strengthen the results section to add such a figure + some analysis into applying the approach over a full winter season (even considering a single IOP).

Reply: As suggested, we have incorporated time-series SWE comparisons in both the two-layer and one-layer SD & SWE comparison results (now Figures 7 and 12).

- Table 2: Was the inclusion of other layering approaches considered? For example, can the model be set up in a way that allows the optimization of the number of snow layers? Only 1- and 2-layer models are presented but show clear improvement one to the other. Does introducing 3-,4- or more layers continue the improvement? Worth adding some additional discussion on this topic

Reply: In the passive microwave study (Pan et al., 2017), we had the option to choose different numbers of layers, but in this paper, it was fixed at two. This decision stems from the observation that the choice of layers can no longer be determined solely by the proximity to the observed signals. Opting for two layers is prudent, particularly because radar tends to overlook the surface layer with fine grain size, as previously discussed in our paper.

We have added the following words in Section 5.2: "Therefore, as a short summary, we recommend considering two layers since radar tends to overlook the surface snow layer of small grain size. Additionally, based on the balance between the current numbers of radar observations and predicted variables, introducing more layers provided little or no improvements in SWE estimation, unless a reliable prior for snow stratigraphy detail becomes available."

**Minor Comments**
- Line 21-22: meaning of '….is not the only promise to obtain..' is unclear. Please revise this sentence for clarity

Reply: Thank you! Please let us know if the following revision will be better: "the role of a precise snow microstructure prior in SWE retrieval may be substitute by a SWE prior from exterior sources."

- Line 58-59: the description of correlation length remains a bit unclear. Is this referring to the distance/length between the solid ice of a single particle to air? Space between particles? It would be useful to make this definition as clear as possible since it is a critical snow microstructure feature in the proposed methodology to get snow depth/SWE

Reply: No. The definition of snow correlation length is similar to the correlation length of rough soil surface, but is that in snow medium in 3 dimensions. We have added the following words in lines 58-62:

"The snow correlation length can considered as the length scale describing the auto-correlation function (ACF) of the ice-air medium, signifying the distance within which this medium can still be considered correlated (Mätzler, 1997). In this study, we specifically estimate the exponential correlation length (Mätzler, 2002) of the snow microstructure. The distinction lies in how the correlation length is determined. While correlation length is fitted from the ACF near the origin, the exponential correlation length is fitted from a longer range of two-point distances in the medium. "

Reference:
Mätzler, C. (1997), Autocorrelation functions of granular media with free arrangement of spheres, spherical shells or ellip- soids. J. Appl. Phys. 81:1509–1517.

• Line 83: there is mention of a semi-empirical variable, but no indication of its importance or what it entails until later in the manuscript. I suggest adding a line describing this variable and its importance here

Reply: Agree. We have added: "The model variable iterated is Q, a semi-empirical parameter to separate the total backscattering into co- and cross-polarization components. "

• Line 89-92: I see the value in this paragraph – but it reads a little awkwardly. Consider revising. Also –    please again specify that the algorithm being used relies on specific MW-radar observations '….radar-based SWE retrieval algorithm..' instead of '…the algorithm..' In its current form, there is no mention of the importance of radar observations in this paragraph

Reply: Agree. We have changed "this algorithm" to "this retrieval algorithm for radar".

• Line 108-110: The authors should provide more information as to why the decision to use VV polarization observations at a 50-deg incidence angle was made.

Reply: We have added the following sentences in lines 119-122:

"We selected this incidence angle to increase the slant penetration path of the radar into the snow medium. However, we avoided choosing an angle that was too large to prevent potential influences from the surrounding environment. We exclusively employed VV polarization, as VH was empirically estimated by MEMLS3&a. It's worth noting that reducing the number of microwave measurements in retrieval typically increases the difficulty of the retrieval process."

• Line 197-199 (& elsewhere): the authors use the form 1+/-0.5 to indicate a mean and standard deviation (as made clear by Table 1) – in my experience this is generally used to show a confidence interval. Please verify that this is the correct approach/format to presenting prior distribution parameters used.

Reply: Thank you for pointing this out. We will change it to N(mean, std) here and in other places.

- Line 220-221: double check numbers here, range was found to be 0.8-1.2, but used 0.1+/- 0.01?
Reply: Yes. The range should be 0.08 and 0.12. Corrections have been made.

- Line 246: double check number – figure shows first 50 iterations
Reply: Thank you! We have changed 20 to 50.

Figure 3: The figure is well put together – but brings up a few questions/suggestions.
- How many iterations does it generally take for the Monte-Carlo simulation to stabilize? Here, it appears there is some adjustment in the first 50 iterations, then the response looks to be similarly distributed for the remaining iterations.
Reply: Yes, the burn-in period can be reduced to 50; however, the total length of chains needs to be extended. This is essential as MCMC aims to establish the posterior distribution, requiring more iterations for its accurate representation. The result of MCMC does not converge to a single point; rather, it converges to a distribution.

- Can you also indicate the mean simulated values on the plot? In addition to the observed ones shown
Reply: Agree. A histogram of simulated backscattering coefficients has been added in this figure.

- Line 278-279: It is inaccurate to refer the all of these as having reduced bias (since one increases its absolute bias) – there is a mention at the end of the paragraph but should be reworded
Reply: Agree. We have revised "reduced" to "changed".

- Figure 7: Great figure! Very clear result
Reply: Thank you!

- Line 301-304: It is not particularly clear to be what is being done to produce the 'fitted' result. It seems the presented model does not reproduce the measurements well – but with some modification it can? Please clarify
Reply: The MCMC-retrieved $p_{ex}$ cannot precisely match the $p_{ex}$ converted from $D_g$ measurements due to the measurement and the parameter conversion uncertainties. However, when we derive a fitted $p_{ex}$ using the same forward model (MEMLS3&a) to match the backscattering observations, it agrees much better with the MCMC-retrieved $p_{ex}$.
To clarify, we have added:
"The fitted $p_{ex}$ was calculated based on the snowpit and soil measurements, an adjustable soil roughness (σ) to match the X-band and a scaling factor for $p_{ex}$ to match the other frequencies. The $p_{ex}$ scaler was constant along the snow profile but varies for different snowpits. "

• Line 331: This seems like a large limitation. If the model does not effectively estimate snow density, how can we expect to get to accurate SWE. Does it rely on changing the depth/layer thickness to get accurate estimates? Please clarify

Reply: No. Actually, adjusting layer thickness and snow microstructure together primarily impacts the estimation of snow depth (SD). Therefore, the accuracy of SWE is benefit from the accuracy of SD estimation and is influenced by the inaccuracy of snow density.

We have added in lines 360:

"This indicates that snow density is difficult to retrieve based on a single low-frequency microwave observation, unless soil liquid water content and soil roughness are provided (Gao et al., 2023), or more observations are provided (Lemmetyinen et al.,2016b)."

References:

Gao, X., Pan, J., Peng, Z., Zhao, T., Bai, Y., Yang, J., Jiang, L., Shi, J., and Husi, L.: Snow Density Retrieval in Quebec Using Space-Borne SMOS Observations, Remote Sens., 15, 1－19, https://doi.org/10.3390/rs15082065, 2023.

Lemmetyinen, J., Schwank, M., Rautiainen, K., Kontu, A., Parkkinen, T., Mätzler, C., Wiesmann, A., Wegmüller, U., Derksen, C., Toose, P., Roy, A., and Pulliainen, J.: Snow density and ground permittivity retrieved from L-band radiometry: Application to experimental data, Remote Sens. Environ., 180, 377－391, https://doi.org/10.1016/j.rse.2016.02.002, 2016.

• Line 354: '…cannot fully correct.', why? Not enough iterations, variables to tune, other?

Reply: We have added: "This occurred because the presence of a surface snow layer was overlooked. This layer generates very small volume scattering, rendering it nearly transparent to radar. Therefore, it is crucial to acknowledge the existence of such a surface layer; otherwise, the total SD will be underestimated. At the same time, because the equivalent one-layer $p_{ex}$ represents that from the bottom layer, it will be larger than the profile-average $p_{ex}$ (from both the $D_g$ measurements and model fittings) (see Fig. 13). "

• Line 370: How is the area of highest probability defined? Should be included somewhere

Reply: It has been revised as:

"The highest probability point at each snow depth was calculated and used to fit a logarithmic equation between SD and $p_{ex}$"

• Line 397: The authors are asserting that this approach could help in SWE estimation at a global scale – still, I think the manuscript lacks discussion into the applicability of this approach at broader scales. Is it feasible to run such an iterative approach over large domains? Can LUT approaches be used using this approach to simulate the snow characteristics based on different combinations of active MW backscatter? These types of questions would be valuable to answer in the discussion before making such claims.

Reply: We believe the term "globally-avaible" is misleading. We will change it to "generic". Additionally, we added a few words at the end of this paper (written in reply to Major point 2) to address your concerns.